# Interactions between temperature and energy supply drive microbial communities in hydrothermal sediment

Lorenzo Lagostina [1], Søs Frandsen[2], Barbara J. MacGregor[3,4], Clemens Glombitza [1,2], Longhui Deng[1], Annika Fiskal [1], Jiaqi Li[1], Mechthild Doll[5], Sonja Geilert [6], Mark Schmidt [6], Florian Scholz [6], Stefano Michele Bernasconi [7], Bo Barker Jørgensen [2], Christian Hensen[6], Andreas Teske [3] & Mark Alexander Lever [1,2✉]

Temperature and bioavailable energy control the distribution of life on Earth, and interact with each other due to the dependency of biological energy requirements on temperature. Here we analyze how temperature-energy interactions structure sediment microbial communities in two hydrothermally active areas of Guaymas Basin. Sites from one area experience advective input of thermogenically produced electron donors by seepage from deeper layers, whereas sites from the other area are diffusion-dominated and electron donor-depleted. In both locations, Archaea dominate at temperatures >45 °C and Bacteria at temperatures <10 °C. Yet, at the phylum level and below, there are clear differences. Hot seep sites have high proportions of typical hydrothermal vent and hot spring taxa. By contrast, high-temperature sites without seepage harbor mainly novel taxa belonging to phyla that are widespread in cold subseafloor sediment. Our results suggest that in hydrothermal sediments temperature determines domain-level dominance, whereas temperature-energy interactions structure microbial communities at the phylum-level and below.

[1] Institute of Biogeochemistry and Pollutant Dynamics, Eidgenössische Technische Hochschule Zürich, Zürich, Switzerland. [2] Department of BioScience, Aarhus University, Center for Geomicrobiology, Aarhus, Denmark. [3] Department of Marine Sciences, University of North Carolina at Chapel Hill, Chapel Hill, NC, USA. [4] Department of Earth and Environmental Sciences, University of Minnesota, Minneapolis, MN, USA. [5] Faculty of Geosciences (FB 05), University of Bremen, Bremen, Germany. [6] GEOMAR Helmholtz Centre for Ocean Research Kiel, Kiel, Germany. [7] Department of Earth Sciences, Eidgenössische Technische Hochschule Zürich, Zürich, Switzerland. ✉email: mark.lever@usys.ethz.ch

Temperature is one of the key variables that control the distribution of life on Earth[1]. Microorganisms isolated from deep-sea hydrothermal environments hold the current upper-temperature record of life at ~122 °C[2]. Theoretical predictions suggest viability at even higher temperatures[3]. Yet, temperature limits may vary according to ecosystem types. Evidence suggests that energy-depleted, diffusion-dominated subseafloor environments in many cases have lower microbial temperature maxima (60–80 °C)[4,5] than environments with advective supplies of external microbial energy substrates. The latter include hydrothermal vents and hot seep sediments (100–105 °C)[6,7], and certain subseafloor sediments (120 °C)[8,9]. This apparently lower temperature limit of diffusion-dominated, strongly energy-limited environments is perhaps related to the exponential relationship between temperature and abiotic biomolecule-damaging reactions, e.g. amino acid racemization and DNA depurination[10]. Accordingly, microorganisms in high-temperature habitats with external energy inputs are better able to compensate for temperature-related increases in maintenance energy requirements than those that inhabit high-temperature habitats where external energy inputs are absent[1,11].

While location-specific interactions between temperature and energy supply appear to set the absolute limits of life in many places, less is known about how interactions between temperature and energy supply influence the community structure of microorganisms. The much higher number of archaeal than bacterial isolates with optimum growth temperatures above 80 °C[12] and the higher reported maximum growth temperatures for Archaea (*Methanopyrus kandleri*, 122 °C)[2] compared to Bacteria (100 °C, *Geothermobacterium ferrireducens*)[13] have been explained with higher thermal stability of archaeal compared to bacterial cell membranes[14]. Yet, pure culture insights do not necessarily translate into the environment. For example, hot hydrothermal vent chimneys and fluids are in some places dominated by Archaea[15,16] and in others by Bacteria[17,18]. A potential explanation is that Archaea cope better with low-energy stress, whereas Bacteria have fitness advantages in energy-rich or unstable environments[12]. While numerous cultivation and cultivation-independent investigations have been done on hydrothermal vent and hot seep sedimentary habitats[18–21], only a small number of studies have investigated the structure of microbial communities in high-temperature subseafloor environments[8,22].

Here we examine the importance of temperature and energy supply (defined as available power) in controlling the microbial abundance and community structure in hydrothermal sediment from two locations in Guaymas Basin, central Gulf of California. Sediments of both locations are dominated by diatomaceous sediment, are anoxic below the sediment surface (≤1 cm), and have wide in situ temperature ranges (≤4 to ≥65 °C). Yet, while sediments of the first area (termed 'seep area' or SA) have high advective inputs of dissolved electron donors (e.g., methane, sulfide, short-chain organic acids (SCOAs)) produced by thermogenic reactions in even hotter, abiotic layers below[23–25], sediment cores we obtained from the larger, more recently discovered second area (termed 'non-seep area' or NSA) are diffusion-dominated and experience no detectable fluid advection[26–28] (for further details on sampling site characteristics, see "General background" and "Site descriptions" in "Methods" section; for general background on Guaymas Basin see Supplementary Methods). Based on sediment cores from five sites in each area, we investigate how relative abundances of Bacteria and Archaea change in relation to temperature, and whether bacterial and archaeal communities differ systematically between hydrothermal sediments that vary in external energy supply. We address these questions by analyzing quantitative and phylogenetic bacterial and archaeal 16S rRNA gene data in relation to in situ temperature profiles, depth gradients of microbial electron donors, electron acceptors, and respiration end products, as well as bulk organic carbon compositional data.

## Results

The results are organized into subsections on in situ temperature profiles, geochemical gradients, and microbial community data. Geochemical data include concentration and isotopic data of dissolved electron acceptors (sulfate, dissolved inorganic carbon (DIC), $\delta^{13}$C-DIC), electron donors (methane, sulfide, SCOAs), and respiration end products (DIC, methane, sulfide), as well as solid-phase organic carbon pools (total organic carbon (TOC), $\delta^{13}$C-TOC, total nitrogen (TN), TOC:TN (C:N)). Microbial community data include bacterial and archaeal 16S rRNA gene copy numbers and bacterial and archaeal community trends. All geochemical and microbiological data are shown in Supplementary Data 1–4.

**Temperature profiles**. The in situ temperatures and temperature gradients differ greatly among sites and hydrothermal areas (Table 1; Fig. 1a, b, 1st column). Certain locations within the SA (Cold Site) and NSA (MUC02, GC13, MUC12) are uniformly cold (~3–5 °C) and thus serve as low-temperature control sites. The fact that Cold Site has no measurable depth-dependent temperature increase suggests that this site, despite being located within the SA, only has minimal hydrothermal fluid seepage. At two sites from the NSA (GC09, GC10), temperatures increase strongly, reaching ~60 °C at 400 cm below the seafloor, with temperature gradients becoming linear below 50 cm. Everest Mound, Orange Mat, and Cathedral Hill in the SA have the steepest temperature gradients (>165 °C m$^{-1}$), reaching >80 °C within 25 cm, whereas Yellow Mat from the SA only reaches ~27 °C at 45 cm. Temperature gradients are near-linear at Everest Mound, Cathedral Hill, and Yellow Mat, and in the top ~15 cm of Orange Mat. Below ~15 cm, the temperatures at Orange Mat are nearly constant.

**Concentrations of methane, sulfate, sulfide, and DIC**. Porewater concentration profiles of methane, sulfate and DIC are consistent with higher microbial activity and higher substrate supplies in hydrothermal seep sediments compared to cold control sites or hydrothermal non-seep sediments.

Independent of temperature, sediments without fluid seepage, i.e. the hydrothermal NSA sites (GC09, GC10) and low-temperature control sites (MUC02, MUC12, GC13, Cold Site), have similar concentration profiles of sulfate, methane, and DIC (Fig. 1a, b, 2nd column). Methane remains at background concentrations (≤0.02 mM), suggesting minimal methane production. DIC concentrations increase with depth by ~1–2 mM relative to seawater values (~2 mM). Sulfate decreases but remains near seawater values (~28 mM) throughout MUC02, MUC12, and the hydrothermal GC10, but drops more clearly toward the bottom of the hydrothermal GC09 (to 26.4 mM) and the cold GC13 (to 23.8 mM). The only minor deviation is Cold Site from the SA. At this site, sulfate and DIC concentrations change more with depth (sulfate drops to 23.6 mM; DIC increases to 6.2 mM), suggesting higher microbial activity relative to all hydrothermal and control sites within the NSA. Consistent with this interpretation sulfide (HS$^-$) concentrations increase strongly with depth at Cold Site (from 2500 to 6200 µM) but not at the NSA sites, where sulfide concentrations remain much lower (0–52 µM (Supplementary Fig. 1). Furthermore, $\delta^{13}$C-DIC decreases with sediment depth at Cold Site (from −3.3‰ to −10.3‰), suggesting strong input of DIC from organic carbon mineralization (Supplementary Fig. 2). By contrast, $\delta^{13}$C-DIC

**Table 1 Overview of all sampling sites.**

| Locations & sites | Latitude (N) | Longitude (W) | Water depth (m) | Depths sampled (mbsf) | $T$ gradient (°C m$^{-1}$) | $T_{max}$ (°C) | Reference |
|---|---|---|---|---|---|---|---|
| *Non-seep area (NSA)* | | | | | | | |
| MUC02 (St. 15) (Ctrl) | 27°26.925′ | 111°29.926′ | 1846 | 0–0.38 | 0.1 | ≤4 | 27 |
| GC13 (St. 9) (Ctrl) | 27°28.193′ | 111°28.365′ | 1838 | 0–4.76 | 0.16 | ≤4 | 27 |
| MUC12 (St. 40) (Ctrl) | 27°24.698′ | 111°23.254′ | 1854 | 0–0.30 | 2.7 | 3.9 | This study |
| GC09 (St. 51) | 27°24.472′ | 111°23.377′ | 1840 | 0–4.87 | 11.4 | 71 | 27 |
| GC10 (St. 58) | 27°24.478′ | 111°23.377′ | 1845 | 0–4.98 | 9.9 | 65 | 27 |
| *Seep area (SA)* | | | | | | | |
| Cold Site (LC3) (Ctrl) | 27°00.542′ | 111°24.489′ | 2011 | 0–0.49 | 2.2 | 4.1 | This study |
| Yellow Mat (Marker 14; LC1) | 27°00.470′ | 111°24.431′ | 2010 | 0–0.45 | 57 | 26.6 | 30 |
| Cathedral Hill (Marker 24; LC10) | 27°00.696′ | 111°24.530′ | 2010 | 0–0.55 | 300 | 118 | 30 |
| Orange Mat (Marker 14; LC15) | 27°00.467′ | 111°24.432′ | 2008 | 0–0.46 | 165 | 89.4 | 30 |
| Everest Mound | 27°00.891′ | 111°24.627′ | ~2000 | 0–0.20 | 490 | 109 | 31 |

Names, geographic coordinates, depth intervals sampled, in situ temperature gradient ($T_{gradient}$), and temperature maximum ($T_{max}$) data for all sites. Uniformly cold control sites are labeled as 'Ctrl' in parentheses after the site names. All $T$ gradients were calculated based on measured temperatures throughout the cored intervals.

remains close to seawater values (~0‰) throughout sediments of all NSA sites (−1.7‰ to −0.2‰).

Compared to all NSA sites and Cold Site, sulfate, methane, and DIC concentrations are more variable at the seep sites Yellow Mat, Cathedral Hill, Orange Mat, and Everest Mound (Fig. 1b, 2nd column). Methane concentrations at Yellow Mat, Cathedral Hill, and Orange Mat are much higher than at the non-seep sites, reaching 3.3, 5.2, and 2.8 mM, respectively (no data from Everest Mound). These high methane concentrations, which can be mainly attributed to the input of thermogenic methane from below[24], almost certainly underestimate in situ concentrations due to outgassing during core retrieval. Sulfate concentrations decrease more strongly with depth than at the NSA sites or Control Site, consistent with previously observed high sulfate-reducing activity[6,7] and advection of sulfate-depleted fluid from below[29]. Nonetheless, sulfate concentrations remain in the millimolar range throughout cores from Yellow and Orange Mat. By contrast, sulfate is below detection (≤0.1 mM) at ≥4.5 cm sediment depth at Everest Mound, and in an intermittent depth interval at Cathedral Hill (~7.5–19.5 cm), below which it increases back to ~6 mM. High, i.e. millimolar, concentrations of sulfide at Orange Mat and Cathedral Hill are consistent with high rates of in situ microbial sulfate reduction and advective input of sulfide from the thermochemical reduction of sulfate in hotter, abiotic layers below (Supplementary Fig. 1). DIC concentrations reach values of >10 mM at Orange Mat, Cathedral Hill, and Yellow Mat (no data from Everest Mound). DIC concentrations fluctuate around 20 mM DIC throughout the core from Cathedral Hill, suggesting high DIC input from deeper layers. C-isotopic values of this DIC are close to those of seawater (~−3‰), suggesting an inorganic source. By contrast, surface sedimentary DIC concentrations at Yellow Mat and Orange Mat are close to seawater values but increase with depth to ~20 and ~14 mM, respectively. Lower δ$^{13}$C-DIC values in surface sediments, which decrease further to values of ~−20‰ to −24‰ at Yellow Mat and −14‰ to −18‰ at Orange Mat within the top 10–20 cm, suggest that most of this DIC comes from the microbial or thermogenic breakdown of organic matter and/or the microbial anaerobic oxidation of methane.

**Trends in dissolved SCOAs across locations.** Porewater concentration profiles of SCOAs are consistent with higher input of reactive organic carbon substrates to hydrothermal seep sediments compared to cold control sites or hydrothermal non-seep sediments.

SCOA concentrations at the cold control sites and hot NSA sites are low, showing no clear depth-related trends, consistent with absence of SCOA input from below and/or biological controlled SCOA concentrations. SCOAs are dominated by acetate (cold MUC02, MUC12, and GC13: 1–3 μM; hydrothermal GCs: 3–6 μM; Cold Site: 1–7 μM), which was detected along with formate, propionate, and lactate (Fig. 2).

By contrast, SCOA concentrations at all hydrothermal seep sites except Orange Mat, increase with depth and temperature, consistent with a thermogenic source below the cored interval. At Yellow Mat, acetate concentrations are already elevated at the seafloor (32 μM) and increase to >100 μM at 20 cm depth. Cathedral Hill has a similar acetate concentration profile, but reaches even higher concentrations (250 μM). At the hottest site, Everest Mound, acetate concentrations increase from ~150 μM at the seafloor to steady concentrations of ~600 μM below 3 cm. Formate concentrations are also (locally) elevated at Yellow Mat (5-8 μM), Cathedral Hill (to 14 μM), and Everest Mound (94-265 μM), and propionate concentrations reach high values at Cathedral Hill (to 21.8 μM) and Everest Mound (to 125 μM). The only exception among the seep sites is Orange Mat, where acetate is only slightly elevated (10–20 μM), and formate and propionate remain at background concentrations. These concentrations suggest that either thermogenic SCOA input from below is low at this site, or SCOA concentrations are biologically controlled throughout the core. Unlike the other three SCOAs, lactate concentrations remain low at all seep sites, apart from one outlier at Cathedral Hill (34.5 cm: 17.3 μM), suggesting that lactate is not a major product of thermogenic organic matter breakdown.

**Trends in solid-phase organic matter pools.** All sites have similar δ$^{13}$C-TOC isotopic compositions, with values ranging from −19‰ to −23‰, consistent with a predominant phytoplankton origin of sedimentary organic carbon (Supplementary Fig. 3). Yet, depth profiles of TOC and TN follow different patterns across the locations (Fig. 3). All cold control sites have similar TOC (~2–4 wt%) and TN contents (~0.3–0.6 wt%), with slight decreases in values from the seafloor downward. Compared to cold controls, GC09 and GC10 have lower TOC and TN contents (TOC: ~0.5–3 wt%; TN: ~0.0–0.3 wt%), in particular in deeper horizons with elevated temperatures. Seep sites within the

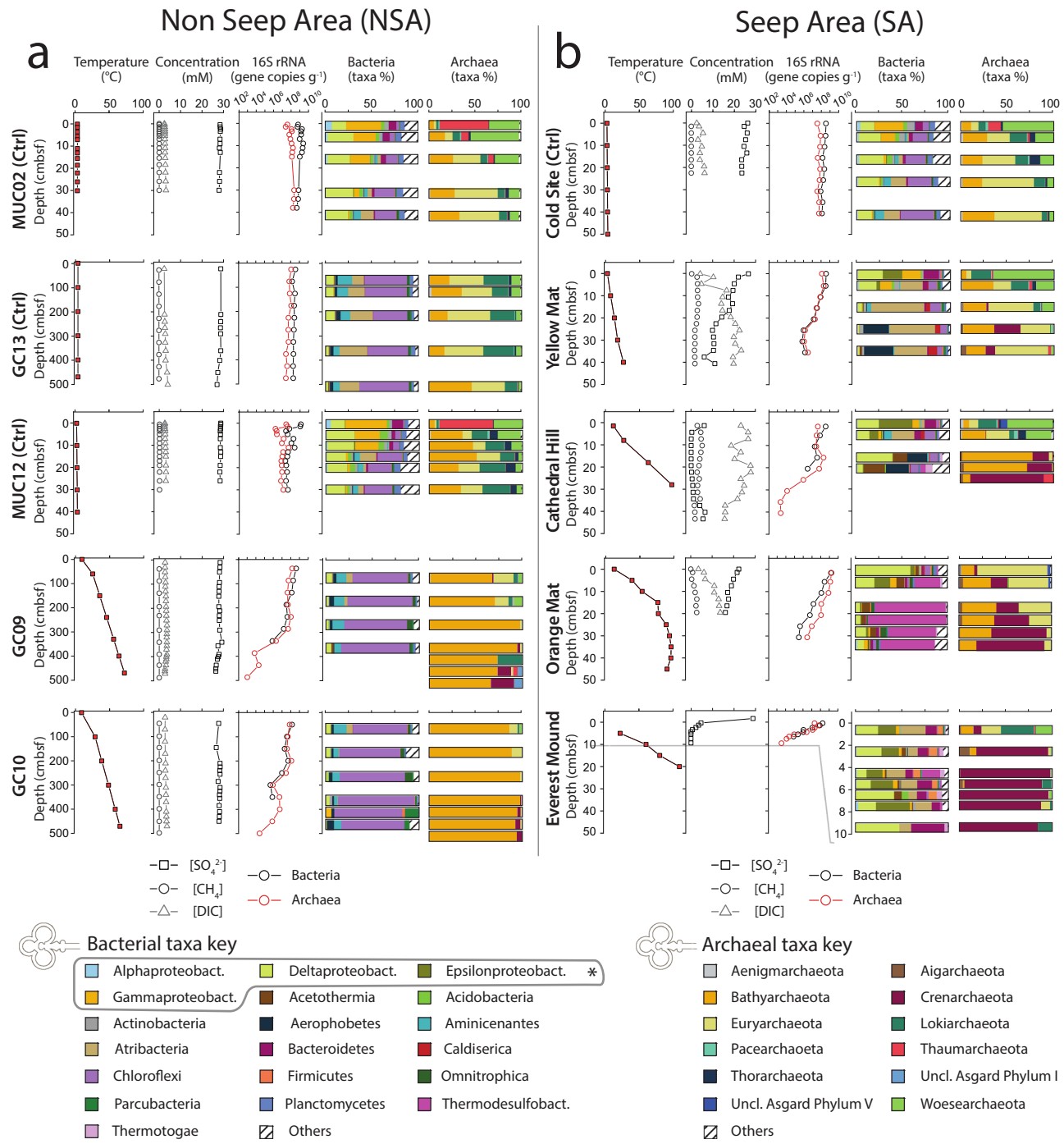

**Fig. 1 Microbial abundance and community structure in relation to temperature and geochemical gradients.** Depth profiles of temperature (1st column), porewater dissolved sulfate, methane, and dissolved inorganic carbon (DIC) concentrations (2nd column), bacterial and archaeal 16S rRNA gene abundances (3rd column), bacterial (4th column) and archaeal community structure (5th column) across the 10 study sites. **a** Sites from the NSA. **b** Sites from the SA. Bacteria and Archaea community structure is shown at the phylum level, except in Proteobacteria, which are displayed at the class level (see asterisk). To improve visibility, we adjusted the depth axis range for bacterial and archaeal communities at Everest Mound, only showing the top 10 cm, where microbial 16S rRNA genes were above detection. Sulfate and methane data from the NSA, except those from MUC12, were previously published[27].

SA have the widest ranges. Seep sites have higher TOC in surface sediment compared to control sites, suggesting net organic carbon assimilation and synthesis by microbial growth. TOC values are 16 wt% at the seafloor of Orange Mat and 6–7 wt% at the seafloor of the other three locations, and then decrease strongly within the top 10 cm, reaching values similar to those of cold sites or hot NSA sites below 10 cm. TN values in surface sediments of seep sites are generally higher than at control sites (~0.7–0.9 wt%),

providing additional evidence of net organic matter synthesis by microbial biomass production, but then decrease steeply to values that are similar to those at hot NSA sites.

As a result of the stable TOC and TN trends, C:N does not change much with depth at the cold locations. Yet, while C:N ranges around 4.4–5.6 at Cold Site, values are considerably higher, around 8.1–10.1, at cold locations within the NSA. By comparison, the hot NSA sites and all seep sites except Orange

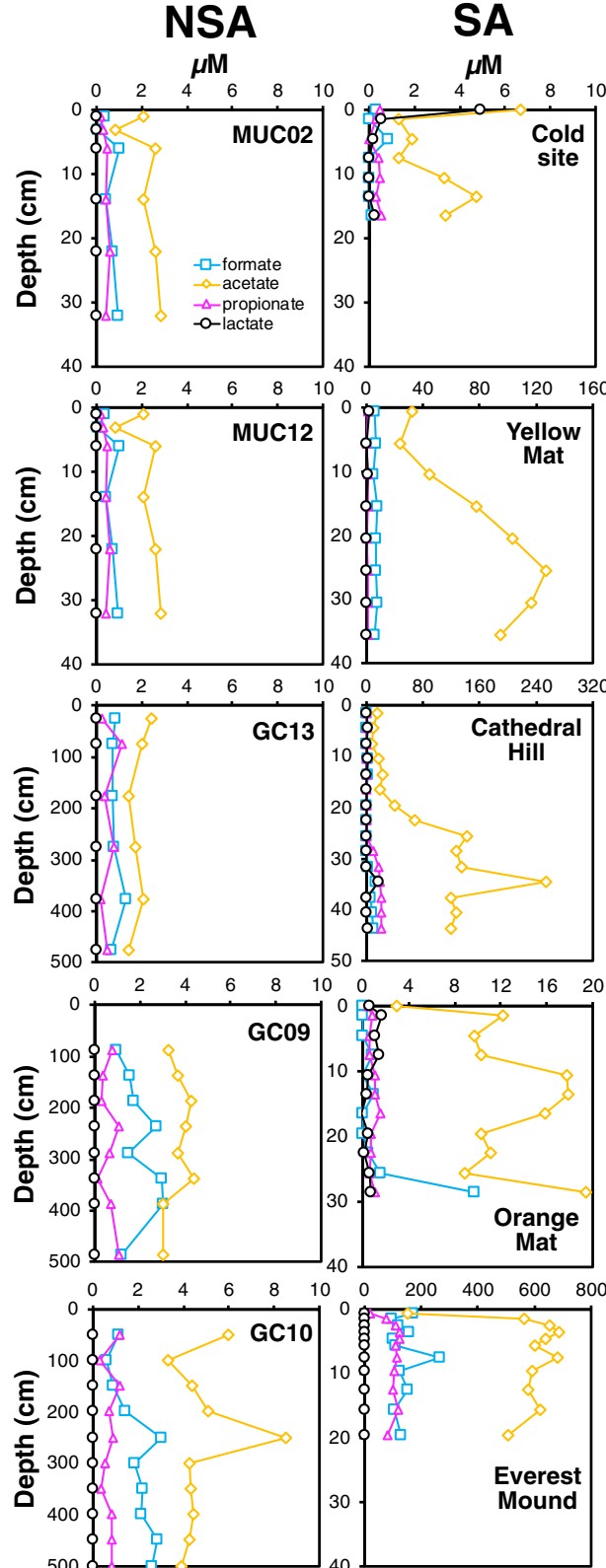

**Fig. 2 Depth profiles of short-chain organic acid (SCOA) concentrations across locations.** Note the differences in concentration ranges on the *x*-axis and depth ranges on the *y*-axis (Cathedral Hill: 0–50 cm; GC13, GC09, and GC10: 0–500 cm; all others: 0–40 cm).

Mat show increases in C:N with increasing temperature and depth. This increase in C:N is modest, from ~8 to 10 at Yellow Mat, and more pronounced at the hotter GC09 (to 15.9), GC10 (to 13.4), Cathedral Hill (to 14.6), and Everest Mound (to 15.7). Orange Mat has the highest C:N ratios (14.8–26.5), and unlike the other sites does not show an increase in C:N with depth.

**General trends in bacterial and archaeal 16S rRNA gene copy numbers**. 16S rRNA gene copy numbers indicate distinct trends in bacterial and archaeal abundances that follow temperature increases with sediment depth (Fig. 1a and b, 3rd column).

At the four cold locations, bacterial and archaeal gene copy numbers are relatively stable with depth (Bacteria: $10^8–10^9 g^{-1}$; Archaea: $10^7–10^8 g^{-1}$). By comparison, gene copy numbers of GC09 and GC10 are in a similar range near the seafloor but decrease strongly with depth. While Archaea are quantifiable throughout both cores to $\leq 10^3$ gene copies $g^{-1}$ sediment, bacterial gene copy numbers are not reliably distinguishable from extraction negative controls ($\sim 1 \times 10^4 g^{-1}$) at temperatures >60 °C. Furthermore, unlike the cold sites, which consistently have higher bacterial gene copy numbers, there is a shift from bacterial to archaeal dominance in gene copy numbers (GC09: at ~50 cm; GC10: at ~150 cm) at both hot NSA sites.

Compared to the hot GCs from the NSA, gene copies decrease over much shorter distances at sites with fluid seepage in the SA. This decrease in gene copy numbers appears related to the magnitude of the temperature increase with depth. At Yellow Mat, which only reaches moderately warm temperatures (27 °C), copy numbers of both domains decrease from $\sim 10^8 g^{-1}$ at the seafloor to $\sim 10^6 g^{-1}$ at the bottom of the core. While Orange Mat, Cathedral Hill, and Everest Mound have similar bacterial and archaeal gene copy numbers to Yellow Mat at the seafloor, these values drop off much more steeply with depth, matching the much steeper temperature increases. At Cathedral Hill and Everest Mound, Bacteria could not be reliably detected below 20 and 7.5 cm, respectively. As the only location, the detection limit of archaeal 16S gene sequences was reached at Everest Mound, at a depth of 9.5 cm.

**Relationships between microbial gene abundances and temperature**. We explored the relationship between 16S rRNA gene copy number and temperature further (Fig. 4a, b). While gene copy numbers of both domains generally decrease with increasing temperature, the shape of this temperature relationship differs between both domains. In bacteria the decrease in gene copy numbers in relation to temperature is nearly linear. By contrast, in Archaea gene copy numbers follow hump-shaped distributions, i.e. they remain stable or only decrease slightly up to a certain temperature threshold, beyond which their copy numbers decrease steeply. This apparent thermal threshold varies between sites, i.e. it is ~85 °C at Orange Mat, ~70 °C at Cathedral Hill, ~50 °C at the NSA sites, and ~20 °C at Everest Mound.

The differences in relationships between bacterial and archaeal gene copy numbers and temperature are reflected in Bacteria-to-Archaea gene copy ratios (Fig. 4c). Bacterial always exceed archaeal gene copies at <10 °C, while archaeal always exceed bacterial gene copies at >45 °C. Between 10 and 45 °C, domain-level gene dominance varies with location. Despite the variability, Bacteria-to-Archaea gene copy ratios follow a highly significant, exponential relationship with temperature ($R^2 = 0.67$, $p < 0.001$ (two-sided Spearman correlation coefficient); Supplementary Fig. 8). This relationship is supported by tests with additional qPCR primer pair combinations (Supplementary Fig. 9). We also investigated Bacteria-to-Archaea gene copy ratio relationships with temperature gradient, sediment depth, concentrations of

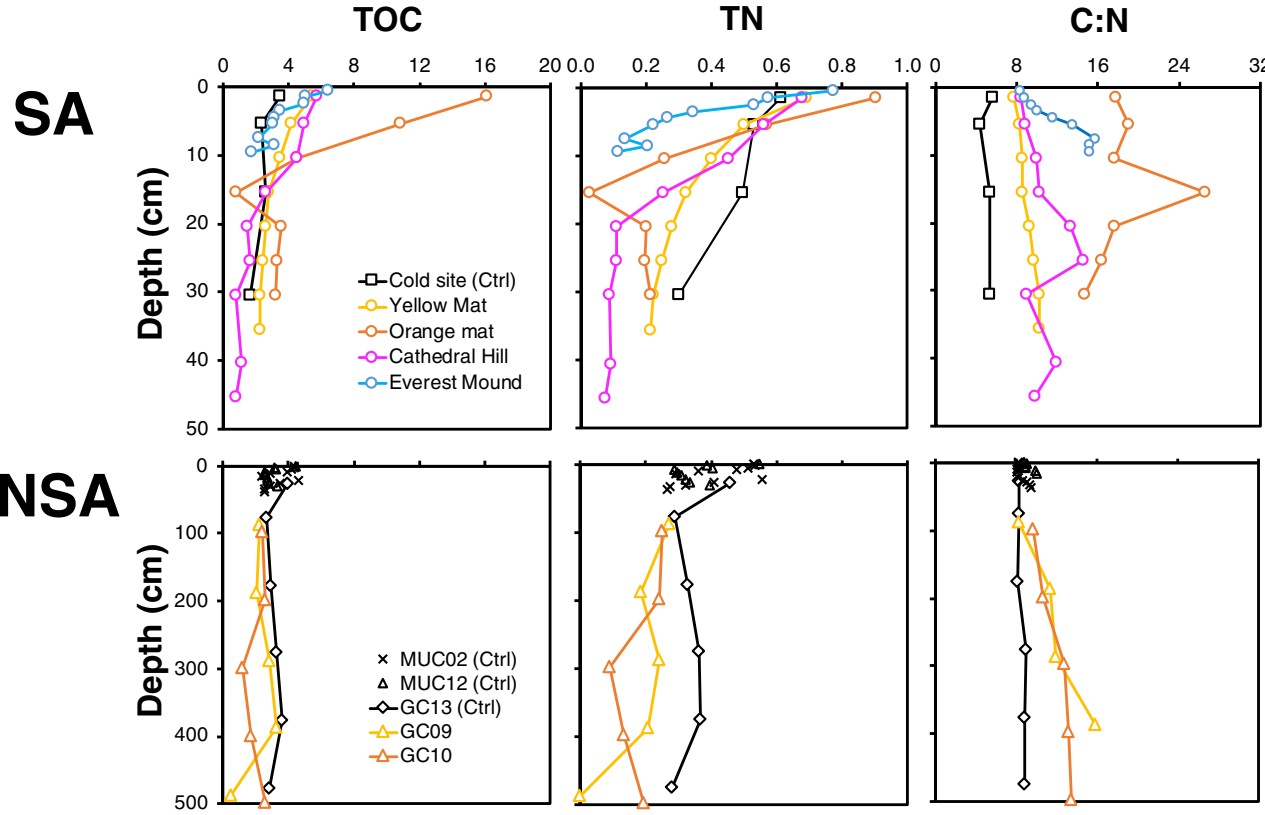

**Fig. 3 Carbon and nitrogen contents of bulk organic matter.** Depth profiles of total organic carbon (TOC), total nitrogen (TN), and TOC:TN (C:N) across all sites.

sulfate, methane, DIC, acetate, formate, and propionate, as well as TOC, TN, C:N, and $\delta^{13}$C-TOC. None are as strong as the correlation with temperature (Supplementary Fig. 8; $R^2_{\text{T-gradient}} = 0.39$; $R^2_{\text{depth}} = 0.02$; $R^2_{\text{sulfate}} = 0.08$; $R^2_{\text{methane}} = 0.30$; $R^2_{\text{DIC}} = 0.30$; $R^2_{\text{acetate}} = 0.13$; $R^2_{\text{formate}} = 0.15$; $R^2_{\text{TOC}} = 0.04$; $R^2_{\text{TN}} = 0.03$; $R^2_{\text{C:N}} = 0.43$; $R^2_{\text{13C-TOC}} = 0.04$).

**Trends in microbial community structure within and between sites.** Microbial communities show clear trends in relation to both temperature and hydrothermal seepage (Fig. 1a and b, 4th and 5th column). This is further confirmed by non-metric multi-dimensional scaling (NMDS) analyses (Fig. 5a, b; Supplementary Fig. 10). Clustering patterns are very similar across the phylum, class, and zero-noise operational taxonomic unit (ZOTU) level within each domain. While cold samples from all locations (and sediment depths) cluster together, bacterial and archaeal communities in sediments with elevated in situ temperatures cluster separately between seep and non-seep locations. In the following sections, we describe major phylogenetic trends across the sampling sites, focusing on phylum-level (Proteobacteria: class-level) trends presented in Fig. 1a, b. For more detailed graphs of dominant microbial taxa at the class level and below, we refer to Supplementary Fig. 4 (Bacteria) and Supplementary Fig. 5 (Archaea). These supplementary figures will be referred to in parentheses when we mention dominant groups below the phylum level in the following text. In addition, due to the high phylogenetic diversity of *Crenarchaeota* and *Bathyarchaeota*, both of which include many unclassified groups, we have extended existing classifications for both phyla based on phylogenetic trees (Fig. 6; for extended classifications see Supplementary Figs. 6 and

7). As a result, we propose 7 new *Bathyarchaeota* subgroups (MCG-24 through MCG-30), as well as several new, order-level subdivisions of *Crenarchaeota*. The new *Crenarchaeota* subdivisions fall into the class *Thermoprotei* ('Deeply branching Thermoprotei'), the Hot Water Crenarchaeote Group I (HWCG I; 'Subseafloor Sediment HWCG I Group' (SSHG)), and the Terrestrial Hot Spring Crenarchaeota (THSC; 'HWCG V' and 'HWCG VI').

The four cold control sites harbor microbial groups that are "typical" of organic-rich bioturbated marine surface sediment (e.g. ref. [30]). Community depth profiles are also similar, despite Cold Site being located in a different part of Guaymas Basin than MUC02, MUC12, and GC13, and despite MUC12 being dominated by metal-rich, hydrothermal vent deposits rather than diatomaceous sediment, which prevails at the other sites. In the short cores (MUC02, MUC12, Cold site), dominance by *Gammaproteobacteria* (mainly BD7-8, *Xanthomonadales*, and unclassified) at the surface (~30–45% of 16S rRNA gene reads) shifts to dominance by *Chloroflexi* (mainly *Dehalococcoidia* consisting of MSBL5, vadinBA26, and unclassified members) below 10 cm (25–40%). Fractions of *Bacteroidetes* (diverse groups), *Acidobacteria* (mainly *Holophagae*) and *Alphaproteobacteria* decrease, while those of *Aminicenantes* and *Atribacteria* increase with depth. *Deltaproteobacteria* (~10–30%; mainly *Desulfobacterales*) and *Planctomycetes* show no depth-related trends. Archaea shift in dominance from *Thaumarchaeota* (Marine Group I) and *Woesearchaeota* in the top 10 cm to *Bathyarchaeota* (mainly C3 (also known as MCG-15) and MCG-17), *Euryarchaeota* (mainly Marine Benthic Group D within *Thermoplasmata*), and *Lokiarchaeota* (mainly Alpha Subgroup) below. Microbial communities in GC13 (shallowest sample:

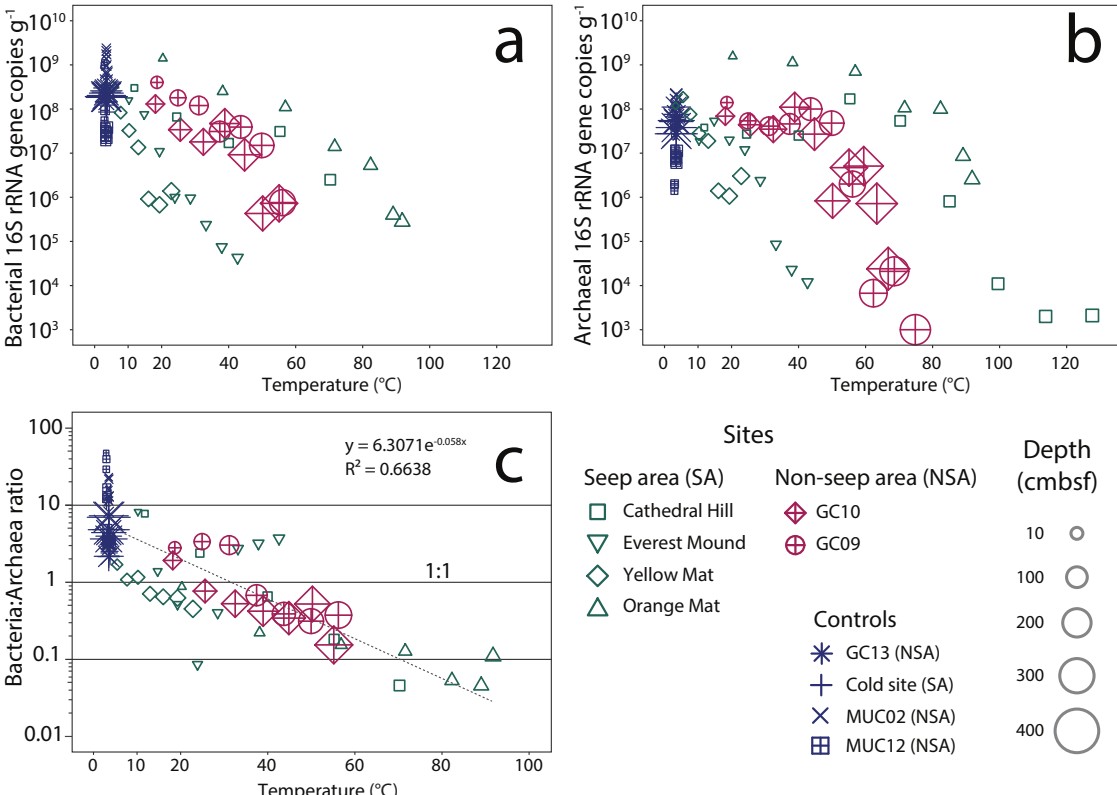

**Fig. 4 Gene copy trends in relation to temperature. a** Bacterial and (**b**) archaeal 16S rRNA gene copy numbers vs. temperature. **c** Bacteria-to-Archaea 16S rRNA gene copy ratios vs. temperature (the exponential function and its coefficient of determination ($R^2$), both calculated in Microsoft Excel, are shown in the graph). Symbol sizes indicate the sediment depth of each sample. Cold control sites from both locations are grouped together in the legend for easier viewing.

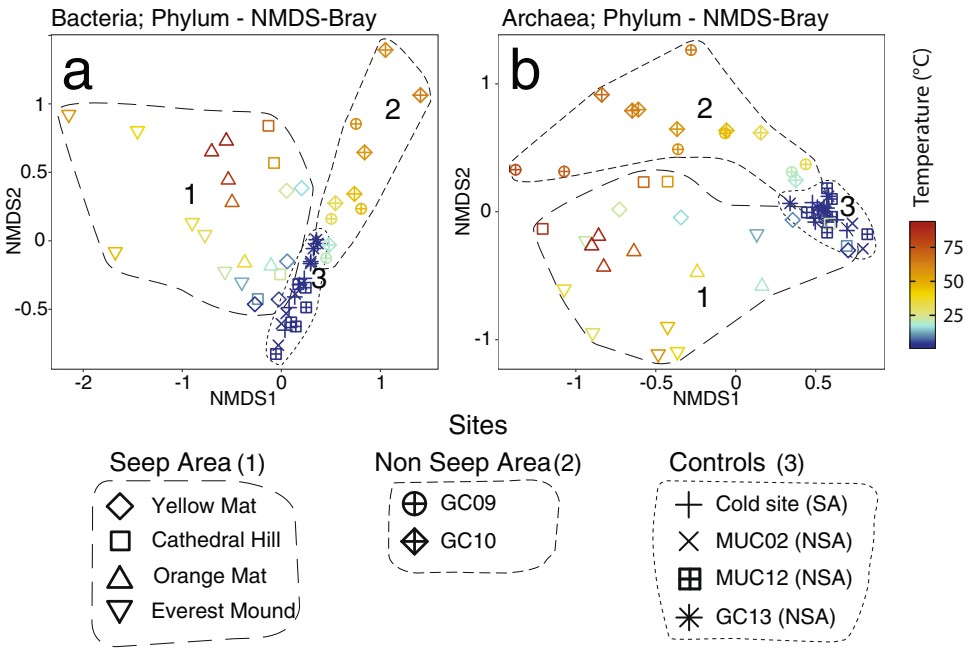

**Fig. 5 Microbial community fingerprints in relation to temperature and locations.** Non-metric multidimensional scaling (NMDS) plots, calculated with Bray–Curtis algorithms, of phylum-level (**a**) bacterial and (**b**) archaeal community structure. Symbol colors indicate sample in situ temperatures. Cold control sites from both locations are grouped together in the legend for easier viewing. For NMDS plots at the class- and Zero-noise operational taxonomic unit (ZOTU)-level, see Supplementary Fig. 10.

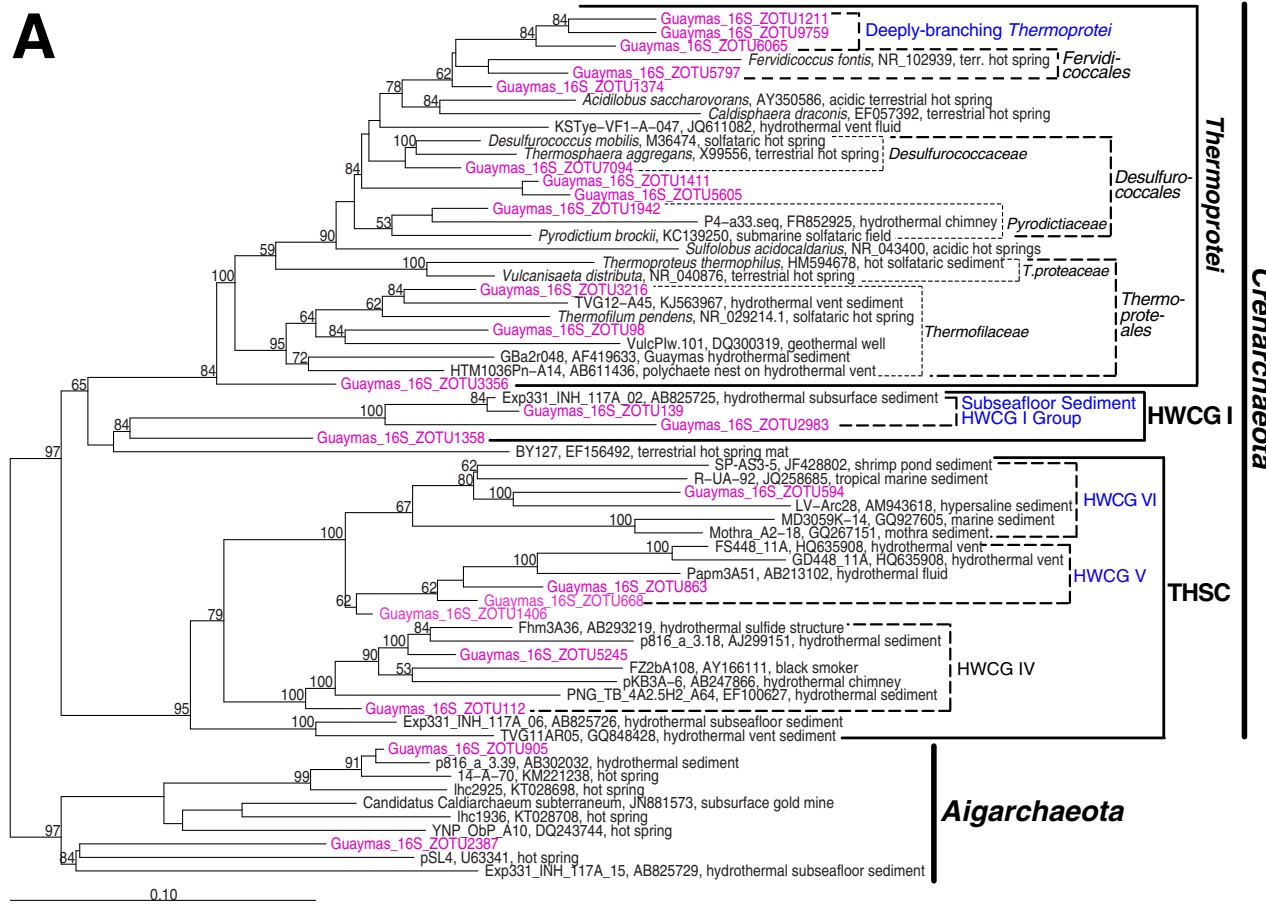

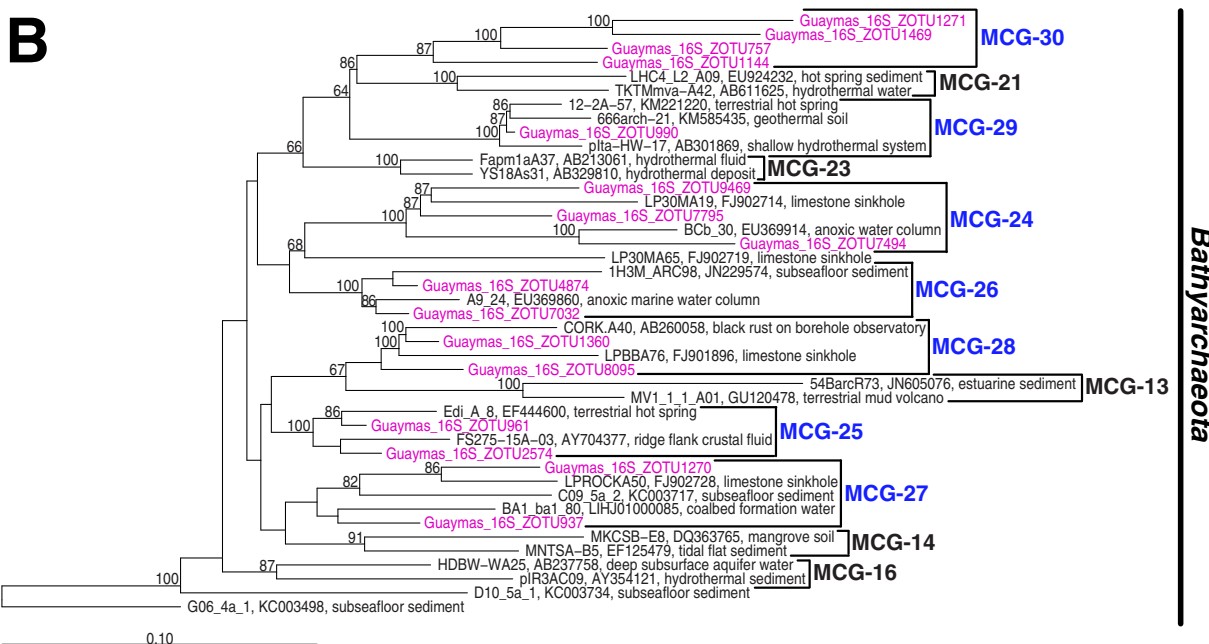

26 cm) resemble those at the bottom of the short cold cores, and change little with depth apart from a decrease in *Deltaproteobacteria*, and a relative increase in *Bathyarchaeota* subgroups MCG-1 and −2 near the bottom.

The two hydrothermal GCs from the NSA differ strongly in microbial community structure from the cold sites, including GC13. Bacteria and Archaea are more clearly dominated by *Chloroflexi* (~65–80% of reads, same major groups of *Dehalococcoidia* as before plus MSB-5B2) and *Bathyarchaeota* (~65–98%), respectively. While the *Chloroflexi* community structure does not change markedly with temperature, there is a clear shift in dominant *Bathyarchaeota* groups. Major groups from control sites, i.e. C3, MCG-1 and −2, only dominate cold surface sediments. MCG-4 dominates temperatures of 40–50 °C

**Fig. 6 Phylogenetic trees of proposed novel archaeal groups.** Phylogenetic trees of (**A**) Cren- and Aigarchaeota, and (**B**) Bathyarchaeota. Selected ZOTUs from this study are shown in pink, newly classified groups in blue. The latter include in (**A**) the Deeply branching Thermoprotei, the Subseafloor Sediment HWCG I Group (SSHG), and the Hot Water Crenarchaeota Groups V and VI (HWCG V and VI), and in (**B**) the MCG-24 through MCG-30. Trees were built in ARB using Neighbor-Joining (Jukes-Cantor Correction) and a 900-bp region column filter that leaves out insertions. Bootstrap trees (1000 replicates) were built using long reads (≥900 bp). Shorter ZOTU reads from this study and ref. [8] were added by ARB Parsimony (for expanded trees, see Supplementary Figs. 6 and 7). The Deeply-branching Thermoprotei were mainly present in hot layers of Orange Mat, whereas the HWCG V and VI occurred at low abundances in hot and cold sediments at various sites within the NSA and SA. Distributions of the SSHG, which was the most abundant of the new Thermoprotei groups, are discussed in the text.

---

between 200 and 300 cm. Further down MCG-16 becomes dominant, and remains dominant to the bottom of GC10, where MCG-3, and the newly classified MCG-27 and -28 (Supplementary Fig. 7) emerge as additional major groups. At GC09, MCG-21, -22, -23, and the newly classified MCG-28 and -29 are the dominant groups below 400 cm. While the hydrothermal GCs have similar percentages of Aminicenantes, Deltaproteobacteria, Omnitrophica, and Aerophobetes compared to GC13, Atribacteria, Euryarchaeota, and Lokiarchaeota have lower percentages in the top 200 cm and are virtually absent below. Instead contributions of poorly known Crenarchaeota increase in deep hot layers. At GC09 these include the class-level Hot Water Crenarchaeota Group IV (HWCG IV) within the class Terrestrial Hot Spring Crenarchaeota (THSC). At GC10 these include the novel, order-level SSHG cluster (Fig. 6; Supplementary Fig. 6). While HWCG IV was previously found in hydrothermal vents and hydrothermal sediments, SSHG was only recovered from an advection-influenced hydrothermal subsurface sediment[8]. In addition, an unknown branch of Asgardarchaeota and an unclassified group of Thaumarchaeota appear below 400 cm at GC09.

Reflecting the variable temperature gradients, geochemical gradients, and advective regimes, microbial communities in hot SA sediments are more diverse and heterogeneous than in hot NSA sediments. Nonetheless, there are shared patterns, that distinguish these sites from all other sites. In surface sediments, nitrifying Marine Group I Thaumarchaeota are nearly absent, and S-cycling Epsilonproteobacteria (Sulfurimonas, Sulfurovum) and Deltaproteobacteria (Desulfobacterales) are dominant Bacteria. In deeper and hotter layers, Aminicenantes are rare and Chloroflexi account for lower percentages. Instead, typical hot spring and hydrothermal vent phyla that have been linked to reductive sulfur cycling and anaerobic organic carbon degradation, i.e. Thermodesulfobacteria, Thermotogae, Acetothermia, and Crenarchaeota (mainly Thermoprotei) dominate these layers.

Despite these similarities, there are also strong differences between seep sites, in particular among bacterial communities in deeper sediment layers. While Atribacteria and Aerophobetes dominate Yellow Mat, Thermodesulfobacteria (T.desulfobacterales) dominate Orange Mat, and S-oxidizing denitrifying Epsilon- (Sulfurimonas, Sulfurovum) and sulfate-reducing Deltaproteobacteria (Desulfobacterales) dominate Everest Mound, no particular group dominates the vertically heterogeneous bacterial community at Cathedral Hill. By contrast, archaeal communities display clear, more temperature than site-related shifts in dominance, from Woese- and Lokiarchaeaota in cold surface sediments to known Bathyarchaeota (C3, MCG-1) in shallow subsurface sediments with moderate temperatures to MCG-22, -23, -28, and -30 in warmer sediments. This is followed by a further community shift toward clear dominance of Crenarchaeota at high temperatures (Thermoprotei dominated by Thermoproteales and Desulfurococcales, at Orange Mat also the novel 'Deeply-Branching Thermoprotei' and HWCG IV). The only exception is Orange Mat, where the top 10–20 cm are dominated by Euryarchaeota linked to anaerobic methanotrophy (ANME-1-Archaea; ANME-1-AT dominates at the surface,

ANME-1-Guaymas (ANME-1-Gba) below), rather than by Woese- and Lokiarchaeota.

## Discussion

Previous studies have postulated that a high-energy supply enables microorganisms in energy-rich habitats to repair damaged biomolecules at higher rates and thus live at higher temperatures than microorganisms in energy-depleted environments[1,4,5,10]. Further studies have proposed that Archaea are better adapted to high temperature and low-energy conditions than Bacteria[12,14]. We study the role of temperature, energy supply, and interactions between temperature and energy supply in driving microbial communities in sediments of two hydrothermal areas within the Guaymas Basin. Study sites from both areas have wide temperature ranges but differ fundamentally in input of external energy substrates (electron donors) via fluid seepage. While seep sediments in the SA experience high advective inputs of dissolved electron donors from below, all sites that we studied in the NSA are diffusion-dominated and lack notable supplies of external electron donors by fluid flow. Our results indicate that temperature, electron donor and hence energy supply, and interactions between both are key drivers of bacterial and archaeal abundance and community structure in hydrothermal sediments.

**Temperature controls absolute abundances at the domain level.** Independent of energy supply or location, bacterial gene abundances decrease in a near-linear fashion with temperature (Figs. 1 and 4a). Archaeal gene abundances in some cases remain stable up to a certain temperature threshold, beyond which they also decrease in a near-linear fashion (Figs. 1 and 4b). Both trends support the notion that increases in energy expenditure due to temperature-driven increases in abiotic biomolecule-damaging reactions, such as amino acid racemization and DNA depurination, lower microbial population size at elevated temperatures. Hereby it seems that the elevated electron donor input to deeper layers of the hydrothermal seep sites does not compensate for temperature-driven increases in cell-specific energy requirements.

Apart from these general patterns, the exact relationships between gene abundances and temperature are site-specific and do not follow differences in energy supply. For instance, the upper-temperature limit of gene detection in the SA ranges from only 40–50 °C at Everest Mound to >90 °C at Orange Mat and Cathedral Hill. Furthermore, in energy-depleted NSA sediments, microbial genes were detected at temperatures of ≥65 °C, and thus at considerably higher temperatures than at Everest Mound with its high inputs of microbial energy substrates, such as SCOAs (Fig. 2).

Variations in temperature gradients caused by fluctuations in fluid flow on time scales of hours to days in the SA[31] may explain the observed variations in temperature maxima. Accordingly, microbial community size, as indicated by gene copy numbers, may in some cases reflect thermal regimes of the past rather than at the time of sampling. For instance, if sediment temperatures

were higher during the days or weeks prior to sampling, then the inferred microbial temperature maximum at Everest Mound could be a strong underestimate of the actual microbial temperature maximum. The reverse reasoning could apply to Cathedral Hill, where archaeal sequences were recovered from sediments with temperatures of ~130 °C (Figs. 1b and 4b). In addition, spatial heterogeneities in thermal regimes could have played a role. For instance, at Orange Mat, temperature differences of 30–40 °C have been reported at 40 cm sediment depth for cores that were horizontally 50 cm apart[31]. Since temperature probes used in the SA were not integrated into core liners but were inserted separately into nearby sediment, measured temperatures may not always accurately reflect temperatures microbial communities within cores were exposed to.

Presumably, measured temperature profiles are more reliable in the NSA, where temperature fluctuations are unlikely due to the absence of fluid flow, and sediment temperatures were measured close to samples on the outside of core liners. We estimate that in these diffusion-controlled, electron donor-depleted sediments the temperature threshold for the archaeal population decrease is 40–50 °C and that archaeal populations are (close to) absent at >80 °C (Figs. 1a and 4b). This estimated upper-temperature limit of ~80 °C in the NSA is in a similar range to temperature limits previously reported for deep hydrocarbon reservoirs[4], continental crust[32], and subsurface sediment with hydrothermal fluid flow[8], but higher than in diffusion-dominated subsurface coal beds[5], and lower than proposed for subsurface sediments in a subduction zone with deep fluid flow[9]. For Bacteria it is not possible to make clear inferences, due to the contamination-related higher detection limit compared to Archaea. If linear extrapolations of gene abundance-temperature trends are an indication, then the temperature limits of Bacteria could be similar to or lower compared to those of Archaea, but this requires additional verification.

**Archaeal dominance at high temperature.** Despite the uncertainties regarding in situ thermal regimes, bacterial gene copy numbers consistently outnumber archaeal gene copy numbers at <10 °C, whereas the opposite is true at >45 °C (Fig. 4c). Even if published 16S rRNA gene copy numbers per cell for the dominant groups present are taken into account, which suggests up to 3-fold lower Bacteria-to-Archaea ratios than at the gene copy level, Bacteria remain clearly dominant in cold sediment and Archaea in warm to hot sediment. These trends are robust across both hydrothermal areas suggesting that relative abundances of Bacteria and Archaea in Guaymas Basin sediment are foremost controlled by temperature.

The dominance of Archaea at >45 °C supports the notion that Archaea are better equipped for life at high temperatures than Bacteria[12]. The reasons for this apparent higher temperature tolerance of Archaea compared to Bacteria, which is based on pure culture studies but has not been consistently observed in the environment, are subject to debate, but are generally attributed to higher heat stability of archaeal membrane lipids (for reviews see refs. [14,33]). Enhanced stability of archaeal membranes at elevated temperature likely lowers membrane permeability, and hence leakage of substrates or membrane potential, and reduces the energetic cost of membrane repair. In both the NSA and SA, these attributes may contribute to a higher energy efficiency and survival rate among Archaea. The fact that Archaea dominate at high temperature in the SA despite temporal variations in temperature suggests that membrane-related archaeal fitness

advantages override reported fitness advantages of Bacteria under more energy-rich or fluctuating environmental conditions[12,34,35].

**Temperature and electron donor supply control microbial community structure.** While temperature likely controls total and relative abundances of Bacteria and Archaea, the combination of temperature and electron donor supply appears to drive microbial community structure at the phylum level and below (Figs. 1 and 2). The high similarity of NMDS clustering patterns across the phylum, class, and ZOTU level (Fig. 5, Supplementary Fig. 10) indicate that distinct combinations of temperature and electron donor supply result in fundamentally different microbial communities. Hereby the main drivers of NMDS clustering patterns are phylum-level differences, with lower taxonomic levels following overarching phylum-level zonation patterns.

Within high-temperature sediment horizons of the SA, known thermophilic Bacteria (*Thermodesulfobacteria, Thermotogae, Caldiserica*) and Archaea (crenarchaeal orders *Thermoproteales* and *Desulfurococcales* within *Thermoprotei*) dominate (Fig. 1, Supplementary Figs. 4 and 5). Strong correlations between relative abundances of these groups with concentrations of electron donors, such as acetate, propionate, and methane (Supplementary Fig. 11) suggest that upward advection of thermogenically produced electron donors drives the distributions of these taxa. This inference is supported by physiological data. Members of *Thermoproteales* and *Desulfurococcales* are metabolically versatile and known to gain energy by fermentation[36,37] and by coupling the oxidation of $H_2$ and diverse soluble organic carbon compounds, including SCOAs, as well as perhaps methane and higher alkanes[21], to the reduction of diverse electron acceptors (e.g., $S^0$, thiosulfate, sulfite, Fe(III))[36,37]. Similarly, *Thermodesulfobacteriaceae* (99% of *Thermodesulfobacteria* reads) couple the oxidation of $H_2$ and SCOAs to the reduction of sulfate, thiosulfate, and sulfite[38], whereas *Caldiserica* uses electrons from labile OM to respire sulfur compounds[39]. The high hydrogen sulfide concentrations and presence of sulfate in most of these high-temperature samples are consistent with the importance of sulfur metabolism in these groups.

The high relative abundances of *Thermotogae, Atribacteria, Aerophobetes, Acetothermia*, and *Bacteroidetes* in high-temperature horizons of the SA also match physiological knowledge on these groups. Genomic and cultivation analyses suggest that members of these bacterial phyla degrade carbohydrates and/or proteins[40–44]. Their occurrence in these layers thus matches the observed heat-driven chemical hydrolysis of insoluble proteins and carbohydrates into their more labile dissolved building blocks in Guaymas Basin hot seep sediment[25]. The strong depletion of N-rich organic matter (Fig. 3) in high-temperature relative to low-temperature horizons supports this interpretation, as it indicates selective breakdown of proteinaceous macromolecules in high-temperature layers.

While past research on the SA has included a strong focus on the high-temperature anaerobic degradation of hydrocarbons, such as thermogenic methane and other short-chain alkanes, our phylogenetic data do not indicate that hydrocarbons are a dominant microbial energy source in hot layers. Additionally, $\delta^{13}$C-TOC values around −22‰ to −20‰ are typical of phytoplankton-derived organic carbon and suggest a minor contribution of deep methane-derived carbon, which has $\delta^{13}$C-values of −39‰ to −52‰ in the SA[45,46]. The same applies to cold surface sediments, where elevated TOC and presence of microbial mats suggest very high in situ microbial biomass production. Taken together these results suggest that microbial growth in the SA, at least at the sites studied, is probably mainly

supported by heat-activation of phytoplankton-derived proteins and carbohydrates rather than by hydrocarbons.

Less is known about the groups that dominate the phylogenetically very different high-temperature horizons of the NSA, none of which include cultured relatives. Unlike hot SA sediments, where members of crenarchaeal *Thermoprotei* are prevalent, hot NSA sediments are dominated by diverse assemblages of *Bathyarchaeota*, including known (subgroups MCG-3, -4, -16, -21, -22, -23) and newly classified subgroups (MCG-27, -28, and -29). Notably, MCG-21, -22, and -23 and close relatives of MCG-28 and -29 were previously recovered from high-temperature settings, whereas sequences closely related to MCG-3, -4, and -27 have been found in cooler subsurface habitats (Supplementary Fig. 7). The energy metabolisms of the newly proposed MCG-27, -28, and -29, as well as MCG-3, are unclear due to the absence of genomic data (MCG-27, -28, -29) and lack of complete biochemical pathways (MCG-3)[47]. Yet, genomes belonging to the remaining five subgroups (MCG-4, -16, -21, -22, -23) have been sequenced and all share complete genetic pathways for carbohydrate fermentation and acetogenesis[47]. Carbohydrate fermentation and acetogenesis are also core metabolic features of subsurface *Dehalococcoidia* and *Aminicenantes*[48–50], members of which dominate bacterial communities in hot NSA layers.

The apparently greater role of acetogenesis distinguishes microbial communities of the NSA from those in the SA, and matches the proposed central importance of acetogenic metabolism in energy-depleted subsurface environments[51]. Furthermore, matching the low concentrations of methane, and the depletion of N-rich/proteinaceous organic matter that was also observed at the NSA (Figs. 1 and 3), the dominant groups in hot NSA layers are not known to degrade hydrocarbons or proteins. Given that fermentative carbohydrate degradation and acetogenesis produce SCOAs, the fate of these metabolites, which remained at low, presumably biologically controlled concentrations, remains unclear (Fig. 2). Low rates of sulfate reduction, which are indicated by slight decreases in sulfate concentrations and low hydrogen sulfide concentrations, are a potential SCOA sink that would match the presence of *Deltaproteobacteria* related to known sulfate reducers (mainly unclassified *Desulfobacterales*). In addition, assimilatory processes, anaerobic oxidation of acetate via a reversal of the acetogenesis pathway, and metal reduction, e.g. by novel and uncultured *Thermoprotei* groups , are plausible sinks for SCOAs.

**Alternative explanations for the observed trends**. The dominance of known thermophilic Bacteria and Archaea at high in situ temperatures in the SA, and the dominance of Archaea over Bacteria at temperatures of >45 °C independent of location, indicate temperature as the main driver of microbial community structure in Guaymas Basin sediment. The important role of energy supply is supported by relationships between microbial community structure and electron donor supply (advection vs. diffusion), electron donor concentrations, and inferred metabolisms of dominant microbial groups. Yet, other variables, e.g. related to dispersal, lithology, or redox conditions could also explain the observed microbial community differences between the two hydrothermal areas. The available evidence suggests that these variables are of likely lesser importance, however.

Despite being located up to tens of kilometers apart in different areas of the basin, all cold control sites have similar phylum-level depth zonations and ZOTU-level compositions (Figs. 1 and 6; Supplementary Figs. 4, 5, and 10). This suggests high dispersal potential, and thus connectivity, between sediments located in different parts of Guaymas Basin. Differences in lithology or metal content are also unlikely drivers. Hemipelagic, diatomaceous sediment dominates most sites of the NSA and SA. Clear vertical shifts in microbial communities are absent in cores where diatomaceous sediments alternate with other lithologies, e.g. iron-rich hydrothermal vent deposits at GC09 and GC10. Strong phylogenetic differences are also absent between hemipelagic, diatomaceous sediment-dominated cold sites (Cold Site, MUC02, GC13) and a cold site with iron-rich hydrothermal vent deposits (MUC12).

Differences in redox conditions are perhaps more likely to explain differences between locations. Indeed, the near-absence of aerobic nitrifying Marine Group I *Thaumarchaeota* and abundance of nitrate-reducing, S-oxidizing *Epsilonproteobacteria* and *Beggiatoaceae* (for more info on latter, see the "Methods" section), indicate a shift from predominantly aerobic to mainly anaerobic chemolithotrophic processes in surface sediments of seep sites. Yet, it is unlikely that differences in redox conditions explain community differences in deeper, high-temperature layers, as in both the SA and NSA sulfate reduction is likely to be the dominant respiration reaction. In the SA, this is evidenced by the dominance of thermophilic S-respiring Bacteria and Archaea, high HS⁻ concentrations, and previous research[6,7]. In cores from the NSA, the substantial abundances of *Desulfobacterales*, decreases in sulfate concentrations, and local accumulations of HS⁻ also point to sulfate reduction as the main terminal respiration reaction in deep, hot layers.

## Conclusions

Our results show that diffusion-dominated, energy-depleted sediments of the NSA are—much like their better-studied counterparts in the advection-dominated SA—a treasure trove of uncharacterized and novel microbial diversity. While temperature alone can explain the dominance of Archaea over Bacteria at elevated temperatures (>45 °C), interactions between temperature and energy supply appear to promote assembly of distinct microbial communities at the phylum level and below. Energy-depleted high-temperature subsurface sediments are dominated by bacterial and archaeal phyla that are also widespread in energy-depleted low-temperature subsurface settings, and likely gain energy from the breakdown of recalcitrant (low-reactivity) carbohydrates and other organic compounds. By contrast, more energy-rich high-temperature seep sediments harbor many groups that are common in hydrothermal vents and terrestrial hot springs. Matching their environmental distributions, many of these groups likely rely on the advective supply of labile dissolved organic compounds from deeper layers as an energy source. The observed domain-level divergence in relation to temperature, and phylum-level divergence in relation to interactions between temperature and energy supply, raise the intriguing possibility that temperature and energy supply have been central drivers of biological evolution since life's early beginnings.

## Methods

**General background**. All sites are listed in Table 1 and shown in Fig. 7. The SA is located in the Southern Trough of Guaymas Basin. All sites were sampled by push cores using the manned submersible *Alvin* during cruises of the *R/V Atlantis* ('Everest Mound': Dive 3204 in 1998; all others in 2009; 'Cathedral Hill': Dive 4565; 'Orange Mat': Dive 4564; 'Yellow Mat': Dive 4563; 'Cold Site': Dive 4567), and are within <1 km of each other. The NSA is located near the Northern Trough and was sampled in 2015 by multicore (MUC) and 6-m gravity corer (GC) during *R/V SONNE* expedition SO241[27,28,52]. The hydrothermal GC09 and GC10 and the cold MUC12 were located in close proximity to a newly discovered vent field[26,27]. Two additional cold sites, MUC02 and GC13, were located ~10 km to the northwest of this vent field.

Accumulations of dissolved manganese ($Mn^{2+}$), iron ($Fe^{2+}$), hydrogen sulfide ($HS^-$) and/or ammonium ($NH_4^+$) to the top cm of sediment indicate shallow $O_2$ depletion and dominance of anaerobic respiration reactions at all sites of the SA and NSA[27,53,54]. With the exception of Cold Site, surface sediments of the SA have

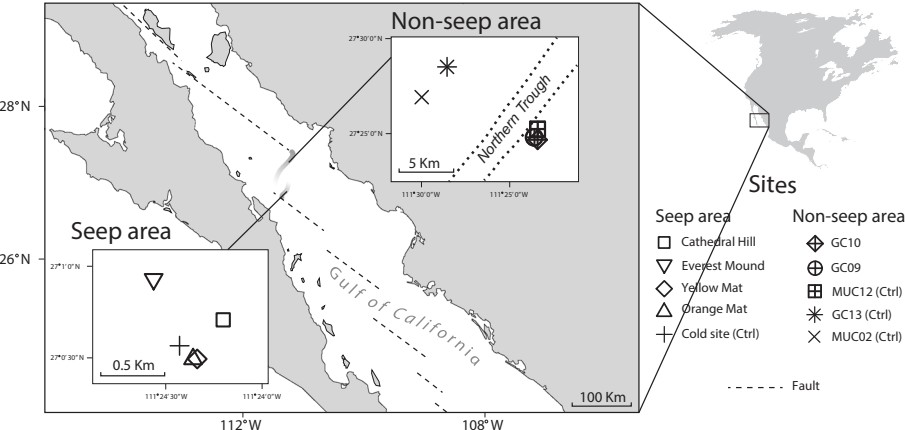

**Fig. 7 Map of study area locations within Guaymas Basin, Gulf of California.** Dashed lines indicate the transform fault, the troughs near both study areas are shown in shaded gray. The much smaller SA was located entirely within the southern Trough. For high-quality seafloor photographs, which were only obtained for the SA and Cold site, we refer to ref. [53]. Map redrawn from www.freevectormaps.com.

clear temperature increases with depth. Microbial electron donors, released by thermal breakdown of relic OM in deeper organic-rich layers and transported to the seafloor by heat-driven fluid circulation[23–25], support high microbial biomass, including microbial mats, at the sediment surface[53]. By contrast, all locations from the NSA, which we examined here, i.e. two geothermally heated and three uniformly cold sites, are diffusion-dominated, and have, based on porewater concentrations and isotopic compositions of inorganic anions and noble gases, not experienced detectable deep hydrothermal fluid advection over the past millennia[27,28]. Organic matter at the two geothermally heated NSA sites becomes progressively degraded throughout the top 4–5 m by a small, but active microbial population, as shown by amino acid compositional and enantiomeric data[11]. The strong depletion in reactive electron donors with increasing sediment depth and in situ temperature is matched by strong decreases in microbial cell counts[11]. For additional geological, geochemical, and microbiological background information on Guaymas Basin, see the Supplementary Methods.

### Site descriptions

*SA*. All sites are dominated by organic-rich diatomaceous sediment. This material is thermally altered at the hot sites called 'Everest Mound', 'Cathedral Hill', and 'Orange Mat', and the temperate site 'Yellow Mat', all of which have hydrothermal fluid seepage combined with clear vertical temperature increases. Hot, shimmering water above the sediment surface was observed at Everest Mound, Cathedral Hill, and Orange Mat. Everest Mound, Orange Mat, and Yellow Mat sediment were visibly oily and had a petroleum smell. While the sediment surface of Cathedral Hill was covered by yellow sulfur precipitates[53], the other sites were covered by *Beggiatoaceae*-dominated microbial mats, ranging from thin white (Everest Mound) to thick orange and white (Orange Mat) to yellow (Yellow Mat). Clear signs of fluid seepage were absent from Cold Site, which is bioturbated and has no mat cover.

*NSA*. All cores are bioturbated and have no mat cover or detectable present-day fluid seepage[27]. GC09 and GC10 sediments consist of intercalated layers of diatomaceous sediment and hydrothermal vent deposits in the upper ~400 cm and transition sharply to metal sulfide-rich, coarse-grained sediment below[27]. MUC12 consists of fine-grained, presumably Fe-rich, hydrothermal vent deposits[52]. MUC02 and GC13 uniformly consist of organic-rich diatomaceous clay.

### Temperature measurements

Temperature profiles at the SA area were measured using heat flow temperature probes that were inserted into the sediment by the *Alvin* submersible (Everest Mound[55]; all other sites[31]). Temperatures in NSA sediment were measured using miniaturized temperature loggers that were directly attached to GCs and MUCs[26].

### Geochemical analyses

*SA*. Sampling for sulfate, methane, and DIC concentration and $\delta^{13}$C-DIC analyses was done as previously described[31]. Porewater for SCOA concentration measurements was sampled by rhizon and measured according to ref. [56] at Cold Site, Cathedral Hill, and Orange Mat. At Yellow Mat and Everest Mound, dissolved SCOAs were sampled by centrifugation and measured according to ref. [57].

*NSA*. Concentrations of sulfate, methane, sulfide, DIC, and SCOAs, and $\delta^{13}$C-DIC were analyzed and measured as previously published[27,57,58]. Porewater for SCOA analyses was sampled by centrifugation and measured according to ref. [57].

TOC, TN, and $\delta^{13}$C-TOC were analyzed as previously described[58].

### DNA extraction

Sediment was sampled using sterile 5-mL syringes with the ends removed. Thick, coherent *Beggiatoaceae* mats, as were present at the sediment surface of the Orange and Yellow Mat sites, were removed prior to sampling. DNA was extracted from 0.2 g of sediment following lysis protocol II as previously published[59] (further details in Supplementary Methods and original reference).

### 16S rRNA gene quantification

Bacterial and archaeal 16S rRNA gene copies were quantified by SYBR-Green-based quantitative PCR (qPCR) on a LightCycler 480 II (Roche Life Science, Penzberg Germany). The Bac908Fmod[59]/Bac1075R[60] and Arch915Fmod[61]/Arch1059R[62] primer combinations were used for Bacteria and Archaea, respectively, following published assays[59] (further details in Supplementary Methods). Samples with on average >3 times higher copy numbers than extraction blanks were included in the manuscript. Likely contaminant sequences that were either overrepresented in extraction blanks or PCR-negative controls or belonged to known laboratory contaminants according to ref. [63,64] were eliminated from the data set.

### 16S rRNA amplicon sequencing and bioinformatic analyses

The archaeal primer pair S-D-Arch-0519-a-A-19[65]/Arch915RRmod[61] and bacterial primer pair S-D-Bact-0341-b-S-17[66]/S-D-Bact-0785-a-A-21[66] were used for amplicon generation and paired-end sequencing (2 × 300 bp) on a MiSeq Personal Sequencer (Illumina, San Diego, USA). Zero-noise operational taxonomic units (ZOTUs; 97% clustering) were generated using UNOISE. Taxonomic assignments were performed using the bacterial SILVA database (SSURef v128) and a manually curated archaeal 16S rRNA gene database in ARB[67], that included recent archaeal metagenome sequences and state-of-the-art taxonomic nomenclature. All ZOTU taxonomic assignments are shown in Supplementary Data 2–4. Further details on sequencing preparations and the bioinformatic pipeline are included in ref. [30] and the Supplementary Methods.

### Statistics and reproducibility

Extraction replicate tests in separate laboratories applying the same extraction method to sediments, but involving different users, subsamples, reaction reagents, and qPCR standards were conducted. The average discrepancy of qPCR-based gene copy numbers was 54% (±32%) for Bacteria and 141% (±57%) for Archaea. These differences do not affect observed trends in Bacteria:Archaea gene copy ratios.

All $\delta^{13}$C-measurements had a reproducibility of ±0.15 per mil (1 SD) based on replicate measurements involving the reference standard Vienna Pee Dee Belemnite (VPDB)[27]. Analytical precisions (1 SD) of concentration measurements were ±0.5 µM for sulfate, ±0.15 mM for DIC, ≤1 µM for methane, ±0.1 µM for acetate, propionate, and lactate[57], and ±0.2 µM for formate[57]. For TOC and TN, the precisions were ±0.1% dry wt.

NMDS analyses were done in R via the Phyloseq package[68], and were calculated using the function "ordinate" (settings: "method = NDMS" and "dist = bray") and then plotted using the function "plot_ordination". Heatmaps were also constructed in R, using relative abundance data of microbial taxa and selected environmental variables. Correlations were calculated using the function "rcorr", specifying "type = spearman" in the package Hmisc version 4.3-0. In the resulting matrices, significant correlations ($p < 0.05$) were visualized using the function corrplot (corrplot version 0.84).

**Reporting summary**. Further information on research design is available in the Nature Research Reporting Summary linked to this article.

## Data availability

ZOTUs of 16S rRNA genes can be retrieved from the GenBank website of the National Center for Biotechnology Information (accession # KDPV00000000). All geochemical data and phylogenetic assignments are provided in Supplementary Data 1–4.

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

## Acknowledgements

We thank the crew of the *R/V Sonne* and *R/V Atlantis* and all scientific participants for their support during the SO241 and AT-15-40 cruises. This work was supported through the German Ministry of Science and Education (MAKS project to C.H.), NSF Biological Oceanography (OCE-0647633 to A.T.), European Research Council Advanced Grant (#294200 to B.B.J.), a Marie-Curie Intra-European Fellowship (#255135), and ETH Zurich internal funding (both to M.A.L.). Additional support was provided by the EU COST Action ES1301 "FLOWS" (https://www.flows-cost.eu).

## Author contributions

M.A.L. designed the study. L.L., S.F., B.J.M., C.G., L.D., A.F., J.L., M.D., S.G., M.S., and F.S. produced all data. S.M.B., B.B.J., C.H., A.T., and M.A.L. contributed analytical tools. L.L. and M.A.L. analyzed the data. L.L. and M.A.L. wrote the paper with input from all co-authors.

## Competing interests

The authors declare no competing interests.
