## [Peer Review File · Communications Biology]

Reviewers' comments:

Reviewer #1 (Remarks to the Author):

Lagostina et al. report an integrated biogeochemical investigation of microbial communities that inhabit seafloor sediments of Guaymas Basin. This site has been studied extensively because of its relatively easy access in the Gulf of California and its scientific value as an experimentally tractable sediment environment that is heavily influenced by hydrothermal processes in the subsurface. The present study is a valuable update to understanding of this system, with new qPCR measurements and Illumina 16S rRNA amplicon sequences. The analyses are expertly performed, and the interdisciplinary synthesis of a broad array of previously published and new biogeochemical data is impressive.

The key result is that archaea are consistently more abundant than bacteria in high-temperature sediments. This is not a surprising result, as it is consistent with previous work at this site and many others, but the results here are unusually convincing and comprehensive thanks to the well-studied site, the high-quality qPCR data, and the comparison of two nearby sites that differ in their temperature and chemical profiles. The bacteria:archaea ratio plot (Fig 4c) is particularly compelling, and I expect that figure to be cited heavily by this community of researchers in the future.

In general, the interpretations of these results are a bit too strong and would benefit from a bit more nuance. For example, although the comparison to the non-seep area (NSA) is an important and valuable component of this study, it is still only an $n=2$ for making the case for a domain-level trend. One group of Archaea (Bathyarchaeota) dominates the non-seep areas while a different group of Archaea (Crenarchaeota) dominates the seep area. This is an exciting and intriguing finding, but the extrapolation from these two data points to a universal domain-level principle should be more careful. Understanding why these groups are dominant at high temperature requires investigating the biology of these specific taxa, and I'm not sure if ascribing these trends to an ancient, domain-level distinction is warranted with the present data or necessarily helpful for understanding this system.

With its extensive description of microbial taxa identified by 16S rRNA amplicon sequencing and its massive phylogenetic trees (in the supplementary materials), the paper reminds me, in a good way, of the classic exploratory studies of novel environments that characterized environmental microbiology in the early days of environmental DNA sequencing. The text is long and descriptive, and much of the discussion is speculative, although the speculations are interesting and well-informed.

The supplementary materials, especially the Excel spreadsheets listing all OTUs and taxonomic classifications, are excellent and very useful companions to the paper. They will be very much appreciated by practitioners in this field.

Additional comments by line number:

52: "according to ecotype" seems to be referring to ecological context, which is not what an "ecotype" is. An ecotype is a species-like unit with a very specific definition based on one theory of speciation.

61: "ecotype" again. I think the authors mean "niche" or "habitat".

64: I'm confused why this sentence is constructed as "while..., less is known...". It seems to me that both halves of the sentence refer to slightly different aspects of the same thing, and they are not in any way opposed to each other.

151 Please clarify that sulfate concentrations decrease rapidly at some sites but not others

167 Please define SCOA here.

265 Please define ZOTU here.

431 Surely there are other potential explanations in addition to membrane stability? It seems too simplistic to discuss this deep biological mystery as though it must be explained by one of the two explanations presented here.

442 The phrase "controlled at the phylum level" seems to slip a little too far down the slope toward implying that selection is acting at the phylum level. All selection is at the species level (or below), of course, so domain-level patterns must be a legacy of ancient evolutionary events or an accident of taxonomic filtering. Either way, nothing is being currently "controlled" at the phylum level today, even if we can observe phylum-level trends today.

554 The grand conclusion "interactions between temperature and energy supply drive microbial community assembly at the phylum level and below" seems too strongly worded because I don't see any reason to focus on the phylum level in particular, as this could be just as easily a genus level pattern that propagates up to the phylum level when plotted that way. In addition, temperature and energy influence the distribution of all life on Earth (as stated in the first sentence of the abstract). The results of this study are consistent with that principle, which is great to see, but the writing style gives the reader the impression that these results have never been seen before.

Reviewer #2 (Remarks to the Author):

The manuscript evaluates the correlation between various geochemical parameters (with a focus on temperature and potential substrate concentrations) and 16S rRNA gene diversity in advection vs diffusion dominated hot sediments. They find temperature correlates best with domain level abundance patterns and that available energy sources may explain community structure at lower taxonomic levels.

The manuscript is well written, clear, and is commended for including some potential caveats in the study and interpretations.

Major comments:

1) I could not find any representation of error or standard deviations on any of the plots. It is hard to interpret what is a significant difference between concentrations versus replicate or measurement error. If error is smaller than the symbol size, please indicate this in figure captions.

2) Similarly, a minimum quantification limit for the qPCR is mentioned, but I could not find this value depicted on the figures or mentioned in the text.

3) Since 16S is always a not a single copy gene, how are differences in copy number between taxa taken into consideration in these comparisons?

4) Since the geochemistry text is mostly descriptive, bringing more of the microbial supplemental figures to the main text would help keep the focus on microbial communities (which the title suggests is the main focus). For example, the description of new taxonomic groups is more significant/impactful than downcore plots of geochemistry and should be brought into the main text.

Table 1 and geochemistry-only figures -> supplement

Figure S1 -> main text, with the addition of panels with seafloor photos that exemplify each site type.

Figure S7 and S8 (or key portions of?) -> main text

Figure S12 -> main text

5) The manuscript could benefit from an overview schematic of dominant processes and taxa

across the compared systems to tie the discussion together.

Minor comments:

- 1) Figure 1 – What is the significance of the line across the bottom, right panel?
- 2) The reader is asked to remember a lot of acronyms specific to this paper/site. It may be worth the extra words in some cases in order to reduce the number of acronyms.
- 3) Figure S9 – color data points by source as in Figure S10.

Reviewer #3 (Remarks to the Author):

The authors compared microbial communities in shallow advection- and diffusion-dominated sediments in Guaymas Basin, Gulf of California. The communities were compared based on 16S rRNA copy number (bacteria vs archaea) and sequence identify (zero-radius operational taxonomic units, ZOTU). The authors also reported geochemical data describing the sediment cores (temperature, temperature gradients, concentrations of inorganic carbon, total organic carbon, total nitrogen, short chain organic acids, sulfate and sulfide). They found that archaea dominate above 45 C and bacteria below 10; at the phylum level high-temperature sites dominated by diffusion have taxa common to cold sediments and hot sites with advection dominating the transport regime have hydrothermal-vent and -spring taxa. The authors conclude that the microbial community structure in Guaymas basin sediments are determined by a combination of temperature and energy availability.

This manuscript is well-written and will be of interest to a broad community of biogeochemists and microbial ecologists. To ensure that these communities, and more, benefit from this manuscript, the authors should consider moving some of the material in the Supplemental Materials section into the main text (see below). My main critical points concern how the authors frame energy and transport and how the discussion is presented. In particular, the authors conclude that energy availability is key to understanding how advective systems can host microorganisms at higher temperatures than diffusion-controlled systems. Yet, they use a rather narrow conceptualization of energy availability being the presence of organic electron donors (and short chain organic acids at that). Unless the microorganisms are all engaged in fermentation (and I see from their data that there are likely fermenters present), then it is the presence of electron acceptors AND donors that will determine the energy availability – if the electrons have nowhere to go, then no energy can be gained. Furthermore, since transport is key to this energy availability, the authors should consider discussing this in terms of the rate of energy availability, or power.

Also, the authors broadly distinguish between Seep Areas and Non-Seep Areas, yet I don't see how these determinations were made. Typically, the Péclet number is used to characterize the relative contributions of advection and diffusion. The authors might not have the data to determine Pe values, but as it stands, it isn't clear how this designation is being made. Also, though some of the sites might be dominated by advection vs diffusion, it's the magnitude of this difference that could be responsible for observed geochemical and microbiological trends. This is also noteworthy with regard to temperature – hot sediments that are dominated by advection will also have faster diffusion rates.

Finally, there is the issue of time. The GC sites have been sampled down to 5 meters whereas some of the other NSAs and all of the SA sites have only been sampled down to between 20-45 cm. Given similar sedimentation rates, this means that the organisms at the bottom of the GC sites have been down there roughly 10 times longer than those at the bottom of the sites where shorter cores were taken. This means that these organisms have endured their entombment in sediments far longer and have likely been subjected to the stresses of low-energy and nutrient deprivation far longer than their shallow counterparts. The authors should consider at least noting this fact and even look for temporal trends in their microbiological data.

Smaller, specific comments

Why is sulfide just considered a respiration end-product and not also an electron donor?
Presumably methane is considered to be both.

Is total organic carbon actually just particular organic carbon (POC), or does it include dissolved OM? This should be clarified.

The section, "Trends in dissolved SCOAs across locations" could use some reorganization. The first sentence needs to encompass what the paragraph will contain, but it starts out with an observation about SCOAs at two sites, then goes into what compounds were found and then a sentence beginning "In contrast..." that doesn't make sense because it does not contrast the preceding sentence.

Is 'TN' total organic nitrogen or total nitrogen? Please specify.

Line 242 The authors could be a bit more specific and scientific than using the phrase 'Hump-shaped' to describe a data trend. Perhaps 'sigmoidal' or 'logistic'?

Define 'ZOTU.' There is a large and growing way to refer to these. And in a few years, who knows which one will be standard.

The authors note that, starting on line 376, "the elevated electron donor input to deeper layers of the hydrothermal seep sites does not compensate for temperature-driven increases in cell-specific energy requirements." However, the authors do not present any values for cell-specific energy requirements at this site, nor do they specify how energy is available from electron donors. Furthermore, they would need to discuss a rate at which this energy is needed since maintenance energies are a rate of energy, not a static amount.

It's great that the authors have a section in the discussion entitled, "Alternative explanations for the observed trends"

The authors note that "Energy-depleted high temperature subsurface sediments are dominated by bacterial and archaeal phyla that are also widespread in energy-depleted low-temperature subsurface settings, and likely gain energy from the breakdown of recalcitrant carbohydrates and other organic compounds." If these compounds are 'recalcitrant' then they wouldn't be broken down. Perhaps they just have a low reactivity under the conditions in which they are found. In many systems complex carbohydrates are broken down readily, with the right combination of organisms and geochemistry. On a related point, if the high temperature, high energy systems are relying on labile organics from deeper, why aren't they being consumed deeper down? Perhaps because the ecosystem of origin doesn't make them labile, but recalcitrant. It is only by being transported to another system that they become labile, a system that has a microorganism that can consume them at an appreciable rate.

In addition to making biomolecular decay more rapid with higher temperatures, abiotic reactions that compete with biologically catalyzed ones also increase with temperature. Furthermore, the energy yield of different reactions changes with temperature. Temperature and energy are more inter-related than the discussion currently allows. The authors should also consider that sulfide is being generated abiotically from hotter sediments via thermochemical sulfate reduction.

The entire paragraph beginning on line 579 should be in the results or discussion or background section! This paragraph answers so many questions that I had about the manuscript up until this point. Why bury this in the Material and Methods section when it is critical background material?

Table 1. The temperature gradient for Everest Mound is 490 °C/m. This might be the case over a very short interval, but it suggests extreme heating, supercritical and/or vapor phase water. A bit misleading.

In figure 1, it looks like nothing much is happening with respects to sulfate reduction, methane

oxidation or organic carbon oxidation in the non-seep areas. I guess the authors don't have nitrate data?

Figure 2. define SCOAs in caption as well as NSA and SA; the caption notes differing scales on the depth (y) axis, but fails to note the same for the concentrations (x axis).

This figure also shows that nothing much is happening in the NSA; the high SCOA concentration deeper at the SA and lower near the SWI makes it look like they are not being utilized at depth, but only near the SWI;

Figure 3 define TOC and TN in caption

TOC profiles make it look like nothing is happening at NSAs, but something at SAs

Figure 4

I'm assuming that the Archaeal 16S rRNA gene copy numbers are the same scale as bacterial, but this should be explicit since the scales change on other plots; upper and lower case letters don't match in caption and plots (a, A)

Figure 5

Spell out NMDS in caption; same with ZOTU

Are the seep areas showing the community structure they are because they are more energy rich, or because microbes from elsewhere are being transported into this area?

The 'Supplementary background on study area' in the Supplementary Text is excellent. Consider including this in the main document for context and background.

Perhaps figure S1 should also be in the main body of the text.

^{13}C seems to confirm that nothing is happening in NSA sediments, w.r.t organic carbon

Figure S12 (heat map showing correlation between microbial groups of geochemical variables) should also be considered for the main text.

Referee expertise:

Referee #1: microbial ecology, metagenomics and astrobiology

Referee #2: Geomicrobiology, environmental microbiology, deep marine microbiology

Referee #3: bioenergetics, geochemistry, thermodynamics and organic matter

Reviewers' comments:

Reviewer #1 (Remarks to the Author):

Lagostina et al. report an integrated biogeochemical investigation of microbial communities that inhabit seafloor sediments of Guaymas Basin. This site has been studied extensively because of its relatively easy access in the Gulf of California and its scientific value as an experimentally tractable sediment environment that is heavily influenced by hydrothermal processes in the subsurface. The present study is a valuable update to understanding of this system, with new qPCR measurements and Illumina 16S rRNA amplicon sequences. The analyses are expertly performed, and the interdisciplinary synthesis of a broad array of previously published and new biogeochemical data is impressive.

The key result is that archaea are consistently more abundant than bacteria in high-temperature sediments. This is not a surprising result, as it is consistent with previous work at this site and many others, but the results here are unusually convincing and comprehensive thanks to the well-studied site, the high-quality qPCR data, and the comparison of two nearby sites that differ in their temperature and chemical profiles. The bacteria:archaea ratio plot (Fig 4c) is particularly compelling, and I expect that figure to be cited heavily by this community of researchers in the future.

Thank you very much for these positive comments regarding our manuscript.

In general, the interpretations of these results are a bit too strong and would benefit from a bit more nuance. For example, although the comparison to the non-seep area (NSA) is an important and valuable component of this study, it is still only an $n=2$ for making the case for a domain-level trend. One group of Archaea (Bathyarchaeota) dominates the non-seep areas while a different group of Archaea (Crenarchaeota) dominates the seep area. This is an exciting and intriguing finding, but the extrapolation from these two data points to a universal domain-level principle should be more careful. Understanding why these groups are dominant at high temperature requires investigating the biology of these specific taxa, and I'm not sure if ascribing these trends to an ancient, domain-level distinction is warranted with the present data or necessarily helpful for understanding this system.

Thank you for this important and constructive comment. We have toned down our interpretation in the Conclusions, and several other places in the manuscript (see author reply to last comment by this reviewer).

With its extensive description of microbial taxa identified by 16S rRNA amplicon sequencing

and its massive phylogenetic trees (in the supplementary materials), the paper reminds me, in a good way, of the classic exploratory studies of novel environments that characterized environmental microbiology in the early days of environmental DNA sequencing. The text is long and descriptive, and much of the discussion is speculative, although the speculations are interesting and well-informed.

The supplementary materials, especially the Excel spreadsheets listing all OTUs and taxonomic classifications, are excellent and very useful companions to the paper. They will be very much appreciated by practitioners in this field.

Thank you again for this overall positive evaluation of our study.

Additional comments by line number:

52: "according to ecotype" seems to be referring to ecological context, which is not what an "ecotype" is. An ecotype is a species-like unit with a very specific definition based on one theory of speciation.

61: "ecotype" again. I think the authors mean "niche" or "habitat".

Thank you for catching this mistake. We have replaced "ecotype" with "ecosystem types" (L. 52/53) or simply "habitats" (L. 61 and 63).

64: I'm confused why this sentence is constructed as "while...., less is known...". It seems to me that both halves of the sentence refer to slightly different aspects of the same thing, and they are not in any way opposed to each other.

The first part of the sentence refers to the general presence/absence of life, the second part to the community structure of microorganisms in those systems where life is present. We have revised the sentence and hope it is more clear now:

"While location-specific interactions between temperature and energy supply appear to set the absolute limits of life in many places, less is known about how interactions between temperature and energy supply influence the community structure of microorganisms."

151 Please clarify that sulfate concentrations decrease rapidly at some sites but not others

Thank you. We have reworded this sentence and added an additional sentence for clarification (L. 158-162).

167 Please define SCOA here.

Done (now L. 175).

265 Please define ZOTU here.

Done (now L. 278-279).

431 Surely there are other potential explanations in addition to membrane stability? It seems too simplistic to discuss this deep biological mystery as though it must be explained by one of the two explanations presented here.

Agreed. We have changed “confer” to “contribute to” (L. 450).

442 The phrase "controlled at the phylum level" seems to slip a little too far down the slope toward implying that selection is acting at the phylum level. All selection is at the species level (or below), of course, so domain-level patterns must be a legacy of ancient evolutionary events or an accident of taxonomic filtering. Either way, nothing is being currently "controlled" at the phylum level today, even if we can observe phylum-level trends today.

Thank you. We agree and have revised the text (L. 459-464):

“Hereby the high similarity of NMDS clustering patterns across the phylum, class, and ZOTU level (Fig. 6, Supplementary Fig. 10) indicate that distinct combinations of temperature and electron donor supply result in fundamentally different microbial communities. Hereby the main drivers of NMDS clustering patterns are phylum-level differences, with lower taxonomic levels following overarching phylum-level zonation patterns.”

554 The grand conclusion "interactions between temperature and energy supply drive microbial community assembly at the phylum level and below" seems too strongly worded because I don't see any reason to focus on the phylum level in particular, as this could be just as easily a genus level pattern that propagates up to the phylum level when plotted that way. In addition, temperature and energy influence the distribution of all life on Earth (as stated in the first sentence of the abstract). The results of this study are consistent with that principle, which is great to see, but the writing style gives the reader the impression that these results have never been seen before.

Thank you very much for this valuable and important comment. We have changed the text accordingly (L. 573-576):

“While temperature alone can explain the dominance of Archaea over Bacteria at elevated temperatures (>45°C), interactions between temperature and energy supply appear to promote assembly of distinct microbial communities at the phylum level and below.”

To further reflect this important point, we have changed formulations in several other places:

L. 44: “show that” → “suggest that”

L. 46: “control” → “structure”

L. 575: “drive” → “appear to promote assembly of distinct microbial communities”

L. 584-588: “The observed domain-level divergence in relation to temperature, and phylum-level divergence in relation to interactions between temperature and energy supply, raise the intriguing possibility that temperature and energy supply have been central drivers of biological evolution since life’s early beginnings.”

Reviewer #2 (Remarks to the Author):

The manuscript evaluates the correlation between various geochemical parameters (with a focus on temperature and potential substrate concentrations) and 16S rRNA gene diversity in advection vs diffusion dominated hot sediments. They find temperature correlates best with domain level abundance patterns and that available energy sources may explain community structure at lower taxonomic levels.

The manuscript is well written, clear, and is commended for including some potential caveats in the study and interpretations.

Thank you for the positive overall evaluation of our manuscript.

Major comments:

1) I could not find any representation of error or standard deviations on any of the plots. It is hard to interpret what is a significant difference between concentrations versus replicate or measurement error. If error is smaller than the symbol size, please indicate this in figure captions.

It is true that most analyses were performed without technical replicate measurements, or based on replicate samples from the same depth, however, the observed depth-related trends within sites, and between sample categories (NSA, SA, controls), are clear and consistent. Within the NSA and control sites, moreover, the observed geochemical and microbiological trends are highly similar, indicating a high degree of reproducibility across sites with similar characteristics. Collectively this suggests (a) high reproducibility and accuracy of all geochemical and molecular biological methods used, and (b) that the geochemical and microbiological variability within the sample depths analyzed was minor compared to the overall trends within each site.

The only exception are extractions of DNA. On sites from the SA we performed two rounds of DNA extractions followed by qPCR, each by a different user. Users used the same extraction method, qPCR assays, and model of real-time PCR cycler, but different qPCR standards, and were located in different countries (Switzerland, Denmark). The qPCR results are not the same, but highly similar. The average discrepancy of bacterial qPCR copy numbers was 54% (+/-32%), whereas the average discrepancy of archaeal qPCR copy numbers was 141% (+/- 57%). These differences are indeed smaller than the symbols in the graphs. The observed discrepancies also do not affect our interpretations regarding Bacteria:Archaea gene copy ratios.

2) Similarly, a minimum quantification limit for the qPCR is mentioned, but I could not find this value depicted on the figures or mentioned in the text.

Thank you for pointing out this missing information. We included all samples that had >3 times higher qPCR values than corresponding extraction blanks in the sequence data set. In addition, we removed likely contaminant sequences that were overrepresented in extraction blanks or previously identified as common laboratory contaminants. We have added the following sentences, and two new references, to the main manuscript (L. 663-667):

“Samples with on average >3 times higher copy numbers than extraction blanks were included in the manuscript. Likely contaminant sequences that were either overrepresented in extraction blanks or PCR negative controls or belonged to known laboratory contaminants according to references 66 and 67 were eliminated from the data set.”

3) Since 16S is always a not a single copy gene, how are differences in copy number between taxa taken into consideration in these comparisons?

Thank you for this comment. 16S rRNA gene copy numbers are currently not being taken into consideration, since the 16S rRNA gene copy numbers of the vast majority of detected organisms are not known. However, based on public databases on 16S rRNA gene copy numbers of genome-sequenced Bacteria and Archaea, our general conclusions regarding relationships between Bacteria-to-Archaea ratios and temperature that are based on gene copy ratios would not change if we accounted for gene copy numbers per cell.

Using the Ribosomal RNA Database website we checked average 16S rRNA gene copy numbers in various dominant archaeal phyla (*Crenarchaeota* class *Thermoprotei*: 1.0; *Thaumarchaeota*: 1.1; *Euryarchaeota*: 2.0; *Bathyarchaeota*: unfortunately no reliable data, but are most closely related to *Cren-* and *Thaumarchaeota*) vs. (*Chloroflexi* class *Dehalococcoidia*: 1.0; *Thermodesulfobacteria*: 1.1; *Thermotogae*: 1.9; *Alphaproteobacteria*: 2.6; *Gammaproteobacteria*: 6.5; *Deltaproteobacteria*: 3.0; *Epsilonproteobacteria*: 2.6). These data suggest that – compared to ratios of bacterial and archaeal 16S rRNA genes - actual Bacteria:Archaea cell ratios are similar in hot NSA sediment, and lower in hot SA sediment and cold controls (up to approximately threefold considering the relative abundances of major groups). Given the observed large ranges of bacterial and archaeal 16S rRNA gene ratios, differences in gene copy numbers per cell between Bacteria and Archaea cannot change our general conclusion that Bacteria dominate at low temperatures (<10°C), while Archaea dominate at warm to hot temperatures (>45°C).

To clarify this important point, we have added the following sentence to the Discussion (L. 435-438):

“Even if published 16S rRNA gene copy numbers per cell for the dominant groups present are taken into account, which suggest up to 3fold lower Bacteria-to-Archaea ratios than at the gene copy level, Bacteria remain clearly dominant in cold sediment and Archaea in warm to hot sediment.”

4) Since the geochemistry text is mostly descriptive, bringing more of the microbial supplemental figures to the main text would help keep the focus on microbial communities (which the title suggests is the main focus). For example, the description of new taxonomic groups is more significant/impactful than downcore plots of geochemistry and should be brought into the main text.

Thank you for this comment. We have added a new figure (Figure 5) which includes summary phylogenetic trees of the extended, supplementary phylogenetic trees that were used to classify unknown groups (Supplementary Figures 6 and 7). This figure is now mentioned in the text, in which we now more clearly highlight the classifications of new

crenarchaeotal groups (L. 293-296; the new bathyarchaeotal groups were already mentioned in L. 291-293).

Regarding the mentioned geochemical figures, we would prefer to keep these in the main manuscript. These figures provide important clues to the *in situ* energy supply to microorganisms, the degradation state and sources of bulk organic carbon, and the presence/absence of fluid seepage – all of which are strongly under the influence of temperature which is also included. These geochemical and temperature gradients/trends are most likely the drivers behind the observed microbial community patterns. The strong importance of these geochemical and temperature data is reflected in the title, which is “Interactions of temperature and energy supply as drivers of microbial communities in hydrothermal sediment”. Due to the interdisciplinary nature of this manuscript and our readership (which is also reflected in the involvement of authors from diverse disciplines), we think the geochemical and temperature data will be of key interest to many readers and should be included.

Table 1 and geochemistry-only figures -> supplement

See previous comment. We consider *in situ* temperatures and temperature gradients key to understanding the observed microbial community patterns.

Figure S1 -> main text, with the addition of panels with seafloor photos that exemplify each site type.

Done. Figure S1 is now Figure 7 in the main text.

For seafloor photos of the SA and Cold site, we refer to a previous publication (see below). We mention these in the caption of this figure (now Figure 7 in main text).

No publication-quality images were obtained for the NSA sites or any of the sites sampled in 2015 as none of these sites were explored by submarine, ROV, or AUV.

Figure S7 and S8 (or key portions of?) -> main text

Done. Now shown in condensed form in Figure 5 in the main text.

Figure S12 -> main text

This figure (now Fig. S11) is not discussed in detail at the moment. We therefore do not think it deserves placement in the main manuscript.

5) The manuscript could benefit from an overview schematic of dominant processes and taxa across the compared systems to tie the discussion together.

We have thought about this possibility carefully, but decided against it in light of the comments of Reviewer #1, who has asked us to tone down the Discussion. Some readers could perceive such a sketch as an overgeneralization of the observed phylogenetic patterns.

Minor comments:

1) Figure 1 – What is the significance of the line across the bottom, right panel?

This line denotes a depth of 10 cm, below which bacterial and archaeal 16S rRNA genes were no longer detectable. We have kept the depth ranges constant across all SA sites, to allow for a better comparison of depth-related temperature and geochemical trends. In order to illustrate the depth distribution of microbial groups in a way that is clearly visible, we have changed the depth scale for the bar charts of bacterial and archaeal communities in this panel. The gray line indicates this (note the depth range is indicated next to the bar charts).

To clarify this point, we have added a sentence to the Figure 1 caption:

‘To improve visibility, we adjusted the depth axis range for bacterial and archaeal communities at Everest Mound, only showing the top 10 cm, where microbial 16S rRNA genes were above detection.’

2) The reader is asked to remember a lot of acronyms specific to this paper/site. It may be worth the extra words in some cases in order to reduce the number of acronyms.

Thank you. We have cut back on abbreviations, or added extra words, in the captions of Figures 1-5, and eliminated the abbreviation BAR for Bacteria-to-Archaea gene copy ratios from the text.

3) Figure S9 – color data points by source as in Figure S10.

Done (now Fig. S8).

Reviewer #3 (Remarks to the Author):

The authors compared microbial communities in shallow advection- and diffusion-dominated sediments in Guaymas Basin, Gulf of California. The communities were compared based on 16S rRNA copy number (bacteria vs archaea) and sequence identify (zero-radius operational taxonomic units, ZOTU). The authors also reported geochemical data describing the sediment cores (temperature, temperature gradients, concentrations of inorganic carbon, total organic carbon, total nitrogen, short chain organic acids, sulfate and sulfide). They found that archaea dominate above 45 C and bacteria below 10; at the phylum level high-temperature sites dominated by diffusion have taxa common to cold sediments and hot sites with advection dominating the transport regime have hydrothermal-vent and -spring taxa. The authors conclude that the microbial community structure in Guaymas basin sediments are determined by a combination of temperature and energy availability.

This manuscript is well-written and will be of interest to a broad community of biogeochemists and microbial ecologists. To ensure that these communities, and more, benefit from this manuscript, the authors should consider moving some of the material in the Supplemental Materials section into the main text (see below). My main critical points concern how the authors frame energy and transport and how the discussion is presented. In particular, the authors conclude that energy availability is key to understanding how advective systems can host microorganisms at higher temperatures than diffusion-controlled systems. Yet, they use a rather narrow conceptualization of energy availability being the presence of organic electron donors (and short chain organic acids at that). Unless the microorganisms are all engaged in fermentation (and I see from their data that there are likely fermenters present), then it is the presence of electron acceptors AND donors that will determine the energy availability – if the electrons have nowhere to go, then no energy can be gained. Furthermore, since transport is key to this energy availability, the authors should consider discussing this in terms of the rate of energy availability, or power.

We thank this reviewer for the overall positive evaluation of our manuscript and the constructive criticisms. We fully agree that terminal electron-accepting reactions are a necessary part of every (anaerobic) microbial food web. We have tried to more strongly emphasize this point in various parts of the manuscript. With respect to energy availability, we originally used this term in the ecological and microbiological sense, which implies a temporal component. We agree that this term can be misleading, and is inaccurate in its literal sense, and have therefore replaced it with the term ‘energy supply’. We now state early in the manuscript that we mean “available power” (L. 80-81), when we use this term. We refrain from switching completely to the term “available power”, because this term may be confusing to most of our expected microbiological, microbial physiological, and ecological readership, especially those readers who are not part of the “low-energy, deep biosphere scientific community”.

Also, the authors broadly distinguish between Seep Areas and Non-Seep Areas, yet I don’t see how these determinations were made. Typically, the Péclet number is used to characterize the relative contributions of advection and diffusion. The authors might not have the data to determine Pe values, but as it stands, it isn’t clear how this designation is being made. Also, though some of the sites might be dominated by advection vs diffusion, it’s the magnitude of this difference that could be responsible for observed geochemical and microbiological trends. This is also noteworthy with regard to temperature – hot sediments that are dominated by advection will also have faster diffusion rates.

The distinction between Seep Areas and Non-Seep Areas is based on past studies, which provide clear evidence of advection of deep, thermally and diagenetically altered fluids at the SA sites (e.g. references 23-25), and no evidence of significant vertical fluid advection through sediments of the NSA sites (references 27 and 28; this is based on noble gas, Mg, and Li concentrations, and helium and strontium isotopic compositions). We have added additional, clarifying information to the Materials & Methods (L. 610-611). In addition, we now more clearly refer to the Materials & Methods and Supplementary Background sections in the Introduction, so interested readers can more easily find this crucial background information on the NSA and SA (L. 90-92).

Finally, there is the issue of time. The GC sites have been sampled down to 5 meters whereas some of the other NSAs and all of the SA sites have only been sampled down to between 20-45 cm. Given similar sedimentation rates, this means that the organisms at the bottom of the GC sites have been down there roughly 10 times longer than those at the bottom of the sites where shorter cores were taken. This means that these organisms have endured their entombment in sediments far longer and have likely been subjected to the stresses of low-energy and nutrient deprivation far longer than their shallow counterparts. The authors should consider at least noting this fact and even look for temporal trends in their microbiological data.

We agree that the microorganisms in deepest cored sediments of the NSA are exposed to a much more severe form of energy limitation than those in the deepest cored sediments of the SA. This is primarily because organic carbon in these deeper layers at the NSA is immobile and progressively degraded through time, whereas deeper layers in the SA receive high advective inputs of reactive substrates released by thermocatalytic reactions of relic organic matter in deeper layers. As for the content and degradation state of bulk organic matter (Fig. 3), it is interesting to note that deep hot layers of the NSA have very similar characteristics as deep layers of the SA, suggesting that high temperature dramatically speeds up the (biotic or abiotic) degradation of organic matter, and is perhaps a better predictor of organic matter degradation state than sediment age.

As for the organisms that dominate the various locations, we observe a clear depth zonation of dominant bacterial and archaeal communities in relation to temperature at sites that have strong temperature increases. By comparison, microbial communities at our control sites with homogeneous temperature regimes, which include one 5-m long gravity core and 3 short cores, change much less with sediment depth (and thus sediment age; e.g. see ordination maps in Fig. 6 and Supplementary Fig. S10). Similar patterns, i.e. only minor changes in microbial community structure in subsurface sediments below the upper decimeters at sites with stable sedimentation regimes and vertically homogeneous organic carbon sources (in this case: diatomaceous organic matter), are common in subsurface sediments. Thus, effects of time since burial are probably minor compared to the impact of temperature.

As for the notion that (individual) organisms have been surviving starvation for longer at the deepest layers of the NSA sites compared to the deepest layers of the SA sites, this is an interesting point. However, it is unlikely that microbial cells at the study sites are survivors from the original deposition at the seafloor. Estimates of microbial living biomass turnover suggest two to three orders of magnitude higher turnover rates, on the order of days to weeks, in deep, high temperature layers of NSA sites – compared to years to decades in cold subsurface sediment (see ref. 11). Thus the DNA detected in hot SA layers in this study was likely from microorganisms that are separated from ancestors that inhabited surface sediments by many (hundreds? Thousands?) generations. Furthermore the main driver of

microbial biomass turnover rates, and thus number of generations since initial deposition, appears to be temperature, not sediment depth, for the depth intervals sampled in this study.

Smaller, specific comments

Why is sulfide just considered a respiration end-product and not also an electron donor? Presumably methane is considered to be both.

Thank you for catching this oversight. Sulfide is certainly also an important electron donor in this system, as discussed later on. We now list sulfide twice in this context, as an electron donor and respiration end product (L. 104 and 105).

Is total organic carbon actually just particular organic carbon (POC), or does it include dissolved OM? This should be clarified.

It is Total Organic Carbon, as we did not remove the porewater which contains dissolved organic carbon (DOC).

The section, “Trends in dissolved SCOAs across locations” could use some reorganization. The first sentence needs to encompass what the paragraph will contain, but its starts out with an observation about SCOAs at two sites, then goes into what compounds were found and then a sentence beginning “In constrast...” that doesn’t make sense because it does not contrast the preceding sentence.

Thank you. We have added a first general summary sentence (L. 176-178) and split the following paragraph in two.

Is ‘TN’ total organic nitrogen or total nitrogen? Please specify.

It is correctly stated in the text. TN is total nitrogen. It is not possible to remove inorganic nitrogen using standard methods of acidification for TOC/TN analyses – these only remove inorganic carbon. For this reason, C:N – a bulk parameter of organic carbon origin and/or quality – technically always stands for the ratio of TOC:TN. Notably, the inorganic N fraction is usually much smaller, usually negligible, compared to the organic N fraction.

Line 242 The authors could be a bit more specific and scientific than using the phrase ‘Hump-shaped’ to describe a data trend. Perhaps ‘sigmoidal’ or ‘logistic’?

We have checked the literature, and the term hump-shaped is widely used in ecology, molecular biology, and economics. Since neither ‘sigmoidal’ nor ‘logistic’ describe the observed trend, and we moreover explain its meaning in our specific context, we think it is – for lack of a better (understandable) word acceptable to use it here.

Define ‘ZOTU.’ There is a large and growing way to refer the these. And in a few years, who knows which one will be standard.

Done.

The authors note that, starting on line 376, “the elevated electron donor input to deeper layers of the hydrothermal seep sites does not compensate for temperature-driven increases in cell-specific energy requirements.” However, the authors do not present any values for cell-specific energy requirements at this site, nor do they specify how energy is available from

electron donors. Furthermore, they would need to discuss a rate at which this energy is needed since maintenance energies are a rate of energy, not a static amount.

This interpretation is qualitative and based on the fact that we studied two end member, high-temperature environments. While the *in situ* temperatures overlap strongly, both locations differ fundamentally in that one receives high advective inputs of electron donors from below, and the other does not (see earlier comments). We simply do not know the cell-specific energy (power) requirements of the cells present, and it would be highly speculative to attempt calculations, given that we do not know advective velocities at the SA sites (except that they fluctuate AND are spatially variable, i.e. much of the advection may proceed through small conduits/channels, which were in some cases even visible in cores).

Interestingly, even though the concentration profiles of electron donors indicate higher microbial activity at the SA compared to the NSA sites, we do not observe systematically higher microbial populations (as indicated by gene copy numbers) at SA compared to NSA sites. One interpretation is that the enhanced availability of electron donors at SA sites has only a minor influence on microbial population size in hot, deeper layers. This could be because the increased energy demand due to temperature-driven increased biomolecule degradation rates negatively impacts microbial population size, and vastly outweighs the positive effect of enhanced electron donor availability (also see L. 391-394). In addition, fluctuations in temperature and fluid compositions could be a very important limiting factor to microbial community size (L. 402-416).

It would be extremely interesting to obtain quantitative data on advective flow rates and potential fluctuations in electron donor supplies to hot sediments through time in order to obtain a more quantitative estimate of microbial *in situ* energy supply (available power) and cell-specific energy consumption rates (cell-specific power demand). However, these data do not exist and would be extremely difficult to obtain (perhaps laboratory experiments would be more suitable). The fact that we do not observe significantly higher microbial populations in hot sediments with elevated electron donor supplies compared to strongly electron donor-depleted hot sediments (Fig. 4) is, nonetheless an important (albeit qualitative) finding.

It's great that the authors have a section in the discussion entitled, "Alternative explanations for the observed trends"

The authors note that "Energy-depleted high temperature subsurface sediments are dominated by bacterial and archaeal phyla that are also widespread in energy-depleted low-temperature subsurface settings, and likely gain energy from the breakdown of recalcitrant carbohydrates and other organic compounds." If these compounds are 'recalcitrant' then they wouldn't be broken down. Perhaps they just have a low reactivity under the conditions in which they are found. In many systems complex carbohydrates are broken down readily, with the right combination of organisms and geochemistry. On a related point, if the high temperature, high energy systems are relying on labile organics from deeper, why aren't they being consumed deeper down? Perhaps because the ecosystem of origin doesn't make them labile, but recalcitrant. It is only by being transported to another system that they become labile, a system that has microorganism that can consume them at an appreciable rate.

Thanks for this comment. The term "recalcitrant" has different uses in the literature, from meaning "no reactivity" to meaning "(very) low reactivity". To clarify our use of the term, we have inserted "low-reactivity" in parentheses after the first use of 'recalcitrant' (L. 579).

Regarding the point about carbohydrates, it is important to note that there is very little data on carbohydrate compositions in deep sea or subsurface sediments, and none on high-temperature sediments, except in sediment porewater of SA sites, where FT-ICR-MS data indicate high dissolved concentrations (ref. 25).

The widespread claim in the geochemical literature that (even complex) carbohydrates are readily/preferentially degraded in surface sediments is not supported by the limited sedimentary inventories (or the implication of ref. 25), and instead often based on incubation experiments with model polymers or fluorescent polymer analogues (e.g. MUF) involving seawater or surficial sediment samples. These pure polymers have very different reactivities compared to the macromolecular, chemically complexed, adsorbed, or physically shielded forms in which significant portions of natural carbohydrates are likely present in sediment.

In fact, research by the Lever group (Zhu et al. (2020) *Org Geochem*; this is a methods paper) on three multicores from Guaymas Basin (cold control sediments; two depths, i.e. surface sediments and 30 cm depth) indicates that carbohydrate contributions to TOC (range: 16-23%) remain high or even increase with depth in cores, providing no evidence of strong selective degradation of carbohydrates over bulk organic carbon. We observe the same trend based on carbohydrate monomer contents and pyrolysis-GC/MS data in a lake sediment column that spans the entire Holocene (Gajendra *et al.*, *in prep.*). Similar work is planned for these sites in Guaymas Basin...

Regarding the point about electron donor consumption in deeper layers of the SA: there almost certainly is electron donor consumption in deep, hot layers of several sites, as has been shown based on sulfate reduction measurements at many locations (e.g. refs. 6 and 7).

Possible reasons for low carbohydrate consumption rates in these deep, hot horizons compared to shallow cold layers could be (a) much lower microbial population sizes (3-6 orders of magnitude lower), which are likely related to lower carrying capacity at high temperature and stress due to thermal fluctuations (see paragraph starting at L. 402), and (b) absence of high energy electron acceptors, such as O₂ or nitrate, which are present at the cold sediment surface. Finally, (c) high rates of *in situ* microbial activity in cold to moderately warm sediment e.g. fermentation, sulfate reduction, methanogenesis, and anaerobic methane oxidation, generate many of the same electron donors (SCOAs, sulfide, methane) that are also produced by thermochemical reactions in deeper layers. This strong increase in biological electron donor production (and hence contribution) toward the seafloor causes an underestimation of turnover rates of deeply-sourced, thermochemically released electron donors if only concentration profiles are considered.

In addition to making biomolecular decay more rapid with higher temperatures, abiotic reactions that compete with biologically catalyzed ones also increase with temperature. Furthermore, the energy yield of different reactions change with temperature. Temperature and energy are more inter-related than the discussion currently allows. The authors should also consider that sulfide is being generated abiotically from hotter sediments via thermochemical sulfate reduction.

Thank you for these interesting comments.

Thermochemical reactions are indeed very important in hot sediments of Guaymas Basin. We already mention this throughout the manuscript, but have added a sentence that explicitly states this to the Introduction (L. 86-88). To highlight the importance of thermochemical

reduction of sulfate, we have added a sentence in the Results (L. 144-147) and Supplementary Text (L. 42-44). In the Introduction, we moreover refer interested readers to our “Supplementary background on study area” section in the Supplementary Text (L. 90-92 of main text).

The extent to which these thermochemical reactions compete with microbial reactions is a very interesting subject, however, the dynamics of this “competition” are not well understood – at least not for the degradation of organic carbon or the reduction of sulfate.

As for the impact of temperature on Gibbs energies of potential microbial catabolic reactions, these certainly could play a role. We have done some calculations and these show the following trends with increasing temperature (assuming that only temperature increases):

- (1) Sugar or organic acid fermentation, acetate oxidation, AOM with sulfate, as well as sulfate reduction and methanogenesis reactions not involving H₂ as substrates become more exergonic, and thus **energetically more favorable with increasing temperatures**.
- (2) Reactions involving the consumption of H₂, e.g. hydrogenotrophic sulfate reduction, methanogenesis, or acetogenesis, become less exergonic, and are thus **energetically less favorable at higher temperatures**.

The entire paragraph beginning on line 579 should be in the results or discussion or background section! This paragraph answer so many questions that I had about the manuscript up until this point. Why bury this in the Material and Methods section when it is critical background material?

We apologize for the confusion, however, it is customary to place background information on sites into the Materials & Methods section for journals which do not have a designated background section. Since this is a synthesis of already published information we do not think it is acceptable to place it into our Results section. To improve clarity, we now refer to these sections also in the Introduction (L. 90-92). The placement of this information is already referred to at the end of the first paragraph of the Results (L. 108-110).

Table 1. The temperature gradient for Everest Mound is 490 °C/m. This might be the case over a very short interval, but it suggests extreme heating, supercritical and/or vapor phase water. A bit misleading.

Thank you. We have added a sentence to the Table 1 caption “All T gradients were calculated based on measured temperatures throughout the cored intervals.”

In figure 1, it looks like nothing much is happening with respects to sulfate reduction, methane oxidation or organic carbon oxidation in the non-seep areas. I guess the authors don't have nitrate data?

Unfortunately there are no nitrate concentration data, however, the fact that Fe²⁺ (and Mn²⁺) became measurable right below the sediment surface (ref. 55), and in deeper layers also HS⁻ became detectable (Supplementary Figure 1), suggests that nitrate reduction is not a dominant respiration process. The Fe²⁺ concentrations from the NSA sites are consistently low and show scatter, which could be due to a combination of precipitation as FeS and O₂ intrusion during sampling (data not shown for this reason). Mn²⁺ concentrations are consistently around 100 μM (main origin unclear, perhaps mineral dissolution). HS⁻ concentrations are in the low micromolar range (Supplementary Figure 1). Collectively, these results indicate

metal reduction and sulfate reduction as the dominant terminal electron accepting processes.

Figure 2. define SCOAs in caption as well as NSA and SA; the caption notes differing scales on the depth (y) axis, but fails to note the same for the concentrations (x axis).

Thank you – fixed.

This figure also shows that nothing much is happening in the NSA; the high SCOA concentration deeper at the SA and lower near the SWI makes it look like they are not being utilized at depth, but only near the SWI;

Yes, but SCOAs are reaction intermediates with (fairly) high turnover rates. It is common for SCOA concentrations to fluctuate within a narrow range (low micromolar to submicromolar) even in sediments with higher activity. Thus (stable) concentrations of SCOAs do not necessarily indicate inactivity. Previous work on this site confirms that there is indeed significant microbial activity (ref. 11). Also see next comment.

Figure 3 define TOC and TN in caption

TOC profiles make it look like nothing is happening at NSAs, but something at SAs

TOC and TN are now defined in the caption.

Despite the fairly stable TOC and TN profiles, the NSA sites are not inactive. This was previously shown by Møller et al. (2018; ref. 11), who – using diagenetic indicators – demonstrated progressive degradation of organic matter with increasing sediment depth at both NSA sites. Møller et al. furthermore showed that the very small population of microbial cells in deep, hot samples of GC09 and GC10 have biomass turnover rates on the order of days to weeks, which is much higher than at a uniformly cold core located outside the NSA.

We now more explicitly state these findings of Møller et al. in the text (L. 613-614).

Figure 4

I'm assuming that the Archaeal 16S rRNA gene copy numbers are the same scale as bacterial, but this should be explicit since the scales change on other plots; upper and lower case letters don't match in caption and plots (a, A)

Your assumption is correct. We have fixed this figure to now show a y-axis scale on the panel showing archaeal gene copy numbers.

Figure 5

Spell out NMDS in caption; same with ZOTU

Done.

Are the seep areas showing the community structure they are because they are more energy rich, or because microbes from elsewhere are being transported into this area?

Microbial community structure at the seep sites clearly reflects the temperature and geochemical regimes at these sites (Figs. 1, 4, and 6; Supplementary Figs. 10 and 11). Typical cold seep taxa dominate cold sediments at the seafloor, whereas typical hydrothermal vent and hot spring classes dominate hot, deep layers.

A deeper source of these latter groups is unlikely given that microbial populations peter out at high temperatures. This would leave possible lateral transport from seawater through tens of centimeters into sediments as the only option. While localized intrusion of bottom seawater is known for these sites, and may explain the presence of millimolar sulfate concentrations throughout deep hot layers (Orange Mat) or the reappearance of sulfate in deep layers (Cathedral Hill), this sulfate input appears to mainly stimulate growth of typical sediment microorganisms.

For instance, matching the high concentrations of sulfate and methane, at Orange Mat members of the ANME-1 clade, which are involved in the anaerobic oxidation of methane coupled to sulfate reduction, are found in significant percentages throughout (with members of thermophilic ANME-1-Guaymas dominating warm/hot deeper layers). ANME-1 typically do not occur in high percentages in hydrothermal vents, which would be the only plausible significant source of thermophilic microorganisms to bottom water in this system. The same applies to Bathyarchaeota. As for Crenarchaeota, the majority of close relatives to sequences detected in this study were previously detected in hydrothermal sediments – not vents.

Lastly, if seawater intrusion had been the main source of microorganisms in deep hot layers of the SA, then we would have expectedly found dominant, or at least high, percentages of typical seawater Bacteria (e.g. Alpha- and Gammaproteobacteria) and Archaea (e.g. Thaumarchaeota) in these layers, which was not the case.

The ‘Supplementary background on study area’ in the Supplementary Text is excellent. Consider including this in the main document for context and background.

Thank you very much for this kind comment, however, given the concise publication format of this journal, and that a significant percentage of readers will mainly be interested in the microbiological results, we prefer to keep this section in the Supplementary Materials. We have made this section more visible/noticeable, however, by mentioning it already at the end of the Introduction, L. 90-92).

Perhaps figure S1 should also be in the main body of the text.

Thank you, we are now including this figure in the main text (Figure 7).

$\delta^{13}\text{C}$ seems to confirm that nothing is happening in NSA sediments, w.r.t organic carbon

Here we kindly disagree. The carbon isotopic composition of bulk organic carbon usually changes if it consists of a mixture of multiple sources with distinct isotopic compositions, where one source is mineralized or assimilated at a higher rate than the other. Examples are (1) favorable degradation of organic carbon from phytoplankton over OC from terrestrial plants, or differential degradation of terrestrial C3 vs. terrestrial C4 plant OC, or (2) high *in situ* assimilation of C from DIC or methane vs. bulk organic matter of largely detrital origin.

The organic carbon that is deposited to Guaymas Basin is almost exclusively of planktonic, and mainly of diatomaceous origin (as stated in the manuscript). Given the uniformity of the organic carbon sources at the NSA sites (no significant inputs of terrestrial organic carbon or thermogenic methane or other (organic) carbon substrates advected from deeper layers), it is not unexpected that the isotopic composition remains fairly constant. It is also worth noting that most organic carbon is mineralized in the top centimeters of the sediments. Thus the actively cycled fraction accounts for only a very small fraction of the total standing stock of organic carbon (TOC) in these deeper layers.

Figure S12 (heat map showing correlation between microbial groups of geochemical variables) should also be considered for the main text.

Thank you, but given the current length of the manuscript, and that this figure (now Figure S11) is not discussed in detail in the main manuscript, we think it is best kept in the Supplementary Materials.

REVIEWERS' COMMENTS:

Reviewer #1 (Remarks to the Author):

I am satisfied by the authors' responses and revisions, and I support publication.

Reviewer #2 (Remarks to the Author):

It is understandable if replicate analyses could not be conducted and therefore standard deviations are not possible for the geochemical measurements.

However, measurement error/detection limits should still be included. It may be that this information exists in the references used in the methods section, and if so this information should be directly included in the methods section of this manuscript because it is critical to interpreting data - especially since the authors indicated that the geochemistry is likely to be of interest to their readers and an important element of this manuscript.

If this is still impossible to provide, then a statement should be included that replicates and standards were not performed in this study with a statement supporting why this is not considered of concern.

All other comments have been adequately addressed.

Reviewer #3 (Remarks to the Author):

Since I have already reviewed an earlier version of this manuscript, I will skip the normal summary and comments. I think that this manuscript is ready for publication. The only additional comment that I have is that the authors have still not defined all all the acronyms in the supplemental figure captions. I won't hold up anything over this. I suspect that anyone looking at the SI will have read the text and understand the acronyms.

Referee expertise:

Referee #1: microbial ecology, metagenomics and astrobiology

Referee #2: Geomicrobiology, environmental microbiology, deep marine microbiology

Referee #3: bioenergetics, geochemistry, thermodynamics and organic matter

REVIEWERS' COMMENTS:

Reviewer #1 (Remarks to the Author):

I am satisfied by the authors' responses and revisions, and I support publication.

Thank you for the positive assessment and the constructive comments to our earlier manuscript version.

Reviewer #2 (Remarks to the Author):

It is understandable if replicate analyses could not be conducted and therefore standard deviations are not possible for the geochemical measurements.

However, measurement error/detection limits should still be included. It may be that this information exists in the references used in the methods section, and if so this information should be directly included in the methods section of this manuscript because it is critical to interpreting data - especially since the authors indicated that the geochemistry is likely to be of interest to their readers and an important element of this manuscript.

If this is still impossible to provide, then a statement should be included that replicates and standards were not performed in this study with a statement supporting why this is not considered of concern.

All other comments have been adequately addressed.

Thank you very much for these positive comments regarding our manuscript. We have provided information on reproducibility of all geochemical and quantitative PCR data in the new section Statistics and Reproducibility (L. 684-694):

“Extraction replicate tests in separate laboratories applying the same extraction method to sediments, but involving different users, subsamples, reaction reagents, and qPCR standards were conducted. The average discrepancy of qPCR-based gene copy numbers was 54% (+/- 32%) for Bacteria and 141% (+/- 57%) for Archaea. These differences do not affect observed trends in Bacteria:Archaea gene copy ratios.

All $\delta^{13}\text{C}$ -measurements had a reproducibility of ± 0.15 per mil (1 SD) based on replicate measurements involving the reference standard Vienna Pee Dee Belemnite (VPDB)²⁷. Analytical precisions (1 SD) of concentration measurements were $\pm 0.5 \mu\text{M}$ for sulfate, $\pm 0.15 \text{ mM}$ for DIC, $\leq 1 \mu\text{M}$ for methane, $\pm 0.1 \mu\text{M}$ for acetate, propionate, and lactate⁵⁸, and $\pm 0.2 \mu\text{M}$ for formate⁵⁸. For TOC and TN, the precisions were $\pm 0.1\%$ dry wt.”

Reviewer #3 (Remarks to the Author):

Since I have already reviewed an earlier version of this manuscript, I will skip the normal summary and comments. I think that this manuscript is ready for publication. The only additional comment that I have is that the authors have still not defined all all the acronyms in the supplemental figure captions. I won't hold up anything over this. I suspect that anyone looking at the SI will have read the text and understand the acronyms.

Thank you very much for the positive assessment. We have now spelled out all acronyms in the supplementary figure captions.